# General transcription factor from *Escherichia coli* with a distinct mechanism of action

Nikita Vasilyev[1,3], Mengjie M. J. Liu[1,3], Vitaly Epshtein[1], Ilya Shamovsky[1] & Evgeny Nudler [1,2] ✉

Gene expression in *Escherichia coli* is controlled by well-established mechanisms that activate or repress transcription. Here, we identify CedA as an unconventional transcription factor specifically associated with the RNA polymerase (RNAP) $\sigma^{70}$ holoenzyme. Structural and biochemical analysis of CedA bound to RNAP reveal that it bridges distant domains of β and $\sigma^{70}$ subunits to stabilize an open-promoter complex. CedA does so without contacting DNA. We further show that *cedA* is strongly induced in response to amino acid starvation, oxidative stress and aminoglycosides. CedA provides a basal level of tolerance to these clinically relevant antibiotics, as well as to rifampicin and peroxide. Finally, we show that CedA modulates transcription of hundreds of bacterial genes, which explains its pleotropic effect on cell physiology and pathogenesis.

The *Escherichia coli* RNA polymerase (RNAP) core enzyme consisting of five subunits (two identical α, β, β′ and ω) is capable of elongating and growing an RNA chain[1]. At different transcription steps, RNAP core forms transient complexes with various accessory proteins. The initiation step is directed by σ factors responsible for sequence-specific recognition of promoter DNA and strand separation[2,3]. Depending on environmental conditions, transcription initiation is finely tuned by a myriad of sequence-specific transcription factors[4] and nucleoid-associated proteins[5].

Besides its primary function in gene expression, RNAP is intimately involved in other essential cellular processes, such as DNA replication and repair[6,7]. Negative supercoiling of DNA, resulting from active transcription in the vicinity of the origin of replication, promotes the binding of a replication initiation protein DnaA. Furthermore, a potential interaction between *E. coli* CedA and RNAP[8,9] ties transcription to another essential process, cell division[10]. However, the nature of this connection remains poorly understood.

Gene encoding CedA was originally discovered as a multi-copy suppressor of the dnaAcos phenotype[10]. At nonpermissible temperature, mutant DnaA causes an over-initiation of DNA replication

leading to nondividing filamentous bacteria. Expression of *cedA* from a multi-copy plasmid restores the *E. coli* capacity to divide. Based on this observation, CedA was ascribed as an activator of cell division.

*cedA* was found among the genes positively selected in uropathogenic *E. coli* strains[11]. The ability of *E. coli* to form nondividing filaments associated with persistent urinary tract infections[12,13] makes a connection to the original discovery of *cedA* as a suppressor of filamentous phenotypes, implying that CedA plays a role in *E. coli* pathogenesis. Another clinically relevant observation was made in the study of bacterial response to gold nanoparticle-based antibiotics[14,15]: CedA overexpression enhances *E. coli* tolerance to a compound named LAL-32, whereas the inactivation of a chromosomal copy of the gene results in higher sensitivity to this compound[15].

Giving its importance for cell division and clinical relevance, several groups set their efforts to identify the biological role and mechanism of CedA. Owing to its small size, CedA was a good target for the structural analysis by nuclear magnetic resonance[8,16]. The structure of its C-terminal domain seemed similar to known structures of double-stranded DNA binding domains, whereas its N terminus

[1]Department of Biochemistry and Molecular Pharmacology, New York University School of Medicine, New York, NY, USA. [2]Howard Hughes Medical Institute, New York University School of Medicine, New York, NY, USA. [3]These authors contributed equally: Nikita Vasilyev, Mengjie M. J. Liu. ✉e-mail: evgeny.nudler@nyulangone.org

was largely unstructured[16]. Indeed, DNA binding site of CedA was identified as TTTTXXT[T/G] using SELEX[8]. However, the biological function and mechanism of CedA remain unknown, which prompted us to address these questions using a combination of quantitative proteomics, cryo-EM, next-generation sequencing (NGS) and biochemical approaches.

## Results

### CedA binds RNAP holoenzyme in vivo

Our approach to study protein–protein interactions in vivo uses immunoprecipitation of the β′ subunit combined with mass spectrometry (MS)-based analysis of coisolated proteins. We used *E. coli* strains bearing chromosomally 3× FLAG-tagged β′ (RpoC) and CedA to ensure natural levels of protein expression, and used crosslinking with formaldehyde to preserve in vivo composition of protein complexes[17].

Proteomic analysis revealed that nearly 1,000 proteins consistently coisolate with RNAP from exponentially growing cells (Fig. 1a). To identify true interactors, we ranked proteins by their abundance; however, even at a relatively high threshold cut-off (2% top-most abundant), less than half of the proteins were RNAP core subunits or known interactors (Fig. 1b and Supplementary Table 1). Many of the highly abundant proteins were ribosomal proteins and translation factors, which we reasoned could bind to the beads nonspecifically and rank highly due to their sheer abundance in bacteria. At the same time, RNAP-interacting proteins expressed at low levels would automatically rank low. CedA, the abundance of which was measured at 235 copies per cell[18] (approximately 10% that of RpoC), is placed at position 129 in the list of the most abundant proteins coisolated with RpoC (Supplementary Table 1), and would not be considered an RNAP-interacting protein. To mitigate these issues, we introduced two additional metrics. First, we performed the pulldown experiments for bacteria bearing no FLAG-tagged proteins to calculate a specificity rank proportional to the difference in abundance between target and nonspecific pulldowns. Second, we measured the level of proteins in a lysate and calculated the enrichment rank proportional to the difference in the abundance between lysates and a target pulldown. With all the metrics combined, 19 of 20 proteins among the top-most ranking 2% were known RNAP interactors (Fig. 1d and Supplementary Table 2), showing a substantial improvement in sorting out nonspecific candidates. Now, CedA is among top-most ranking proteins at position 10, following RNAP core subunits, sigma factors RpoD and FecI, and the elongation, termination and/or antitermination factors NusA, NusG and SuhB (Supplementary Table 2).

Using this improved procedure, we performed the same measurements for CedA, and when the top-most ranking 2% of the proteins coisolated with CedA were compared to those isolated with RNAP, seven were common between both targets (Fig. 1c and Supplementary Tables 2 and 3). These proteins were RNAP core subunits (RpoA, RpoB, RpoC and RpoZ), CedA, transcription initiation factor σ70 (RpoD) and stringent starvation protein A (SspA). This result implied a direct involvement of CedA with σ70-mediated transcription initiation.

### CedA resides at σ70 promoters

CedA was reported to bind DNA, presumably via its C-terminal domain, which is structurally similar to known double-stranded DNA (dsDNA) binding proteins[16]. The SELEX study[8] identified a consensus binding sequence for CedA as a short T-rich stretch of DNA. These reports, together with our initial observation of CedA affinity to RNAPσ70 holoenzyme, led us to hypothesize that CedA could be a sequence-specific transcription initiation factor. To test this hypothesis, we performed a chromatin immunoprecipitation with sequencing (ChIP–seq) experiment to identify CedA-binding sites on the *E. coli* chromosome.

We isolated RNAP and CedA from formaldehyde-treated bacteria by immunoprecipitation and analyzed the coprecipitated DNA using

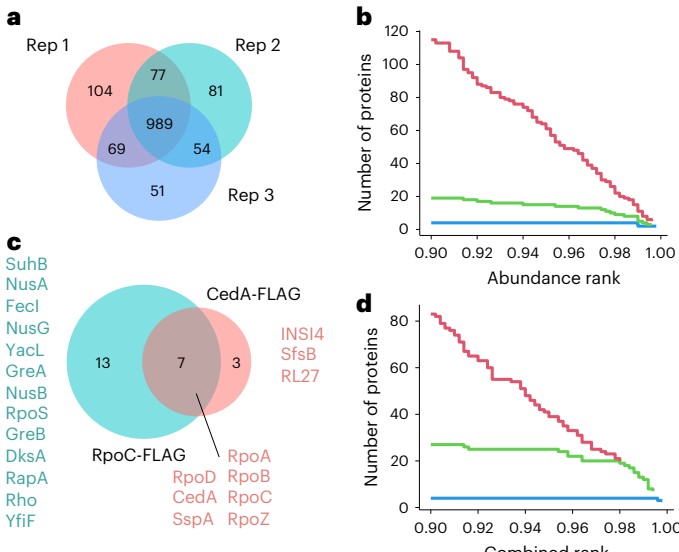

**Fig. 1 | RNAP and CedA interactomics in *E. coli*. a**, Venn diagram showing an overlap of proteins coisolated with RNAP (FLAG-tagged RpoC) in three independent experiments. **b**, Step plot showing a number of proteins identified at a given abundance rank threshold. Plotted are the top 10% of the most abundant (rank 0.9–1.0) proteins. Colors represent RNAP core subunits (blue), known RNAP interactors (green) and other *E. coli* proteins (red). **c**, Venn diagram showing distinct and common proteins coisolated with both RNAP (FLAG-tagged RpoC) and CedA (FLAG-tagged) identified at a 2% combined rank threshold. **d**, Step plot showing a number of proteins identified at a given combined rank threshold (top 10% proteins plotted). Colors are the same as **b**.

high-throughput sequencing. Sequencing depth profiles demonstrate that CedA locates preferentially at promoter regions, whereas the core RNAP signal is distributed along entire transcription units (Fig. 2a). To identify a consensus sequence of a CedA-binding site, we took 20 base pair (bp) long DNA sequences centered around CedA peaks and aligned them using MAFFT v.7.487[19]. The resulting sequence (Fig. 2b) matches the σ70 promoter −10-consensus TATAAT[20]. Further analyses revealed that most identified CedA peaks (301 of 473) are indeed located around −10 region of *E. coli* promoters, and map within the 20-bp distance from known transcription start sites (Fig. 2c). Among the 172 unassigned CedA peaks, 117 were mapped to the intergenic regions, implying their possible association with yet uncharacterized promoters and transcription start sites (Supplementary Table 4).

A combination of high-throughput proteomics and NGS provided strong evidence supporting the involvement of CedA in transcription initiation: CedA binds to RNAPσ70 holoenzyme and is localized around the −10-consensus sequence of σ70 promoters. It associates with promoters of hundreds of genes involved in every essential process including transcription, translation, DNA replication, repair, recombination, transport and metabolism of nucleotides, amino acids, lipids, carbohydrates and cofactors, cell wall synthesis, cell division and others (Supplementary Table 4).

### Structure of CedA bound to the open-promoter complex

To gain further insight into the mechanism of CedA, we used cryo-EM to solve the structure of CedA bound to the RNAPσ70-open-promoter complex.

Purified RNAPσ70 holoenzyme was first incubated with 85-bp DNA representing −60 to +25 positions relative to the transcription start site of the *ssrA* promoter (Fig. 3a), which has the strongest association with CedA, as determined by ChIP–seq (Fig. 2a and Supplementary Table 4). Purified CedA was added to the preformed RNAPσ70·ssrAp

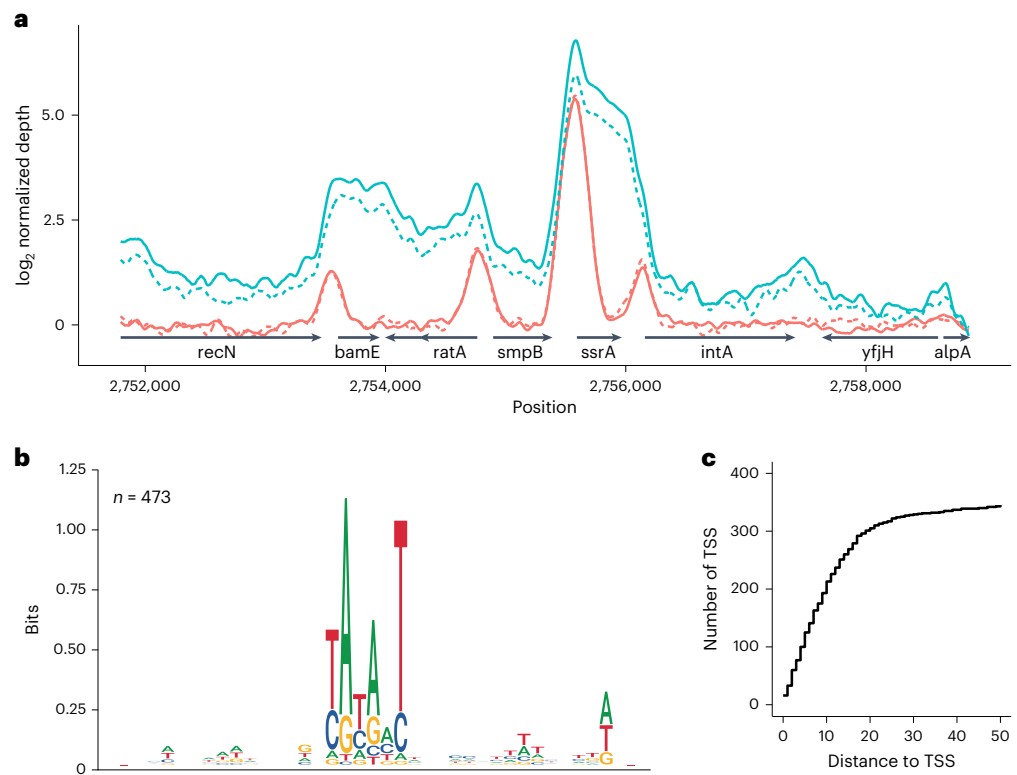

**Fig. 2 | CedA colocalizes with the RNAPσ70 holoenzyme throughout the *E. coli* genome. a**, The smoothed ChIP–seq sequence depth profile of RNAP (FLAG-tagged RpoC, cyan) and CedA (FLAG-tagged CedA, red). Solid and dashed lines represent two replicates of the experiment. **b**, Consensus DNA sequence identified by aligning 20-bp long DNA segments centered around CedA peaks. **c**, Step plot showing a number of transcription start sites (TSS) mapped at various distances from CedA peaks.

and the resulting complex was repurified by gel filtration and used for single-particle cryo-EM analysis. A structure of the whole complex was determined at the overall resolution of 2.76 Å (Table 1 and Extended Data Fig. 1), which let us to build the CedA model de novo (Extended Data Fig. 2).

The general architecture of RNAP in the complex (Fig. 3b) is similar to the known structure of a transcription initiation complex[21]. We identified CedA adopting a ladle-like shape and binding the RNAP at the tip of a β-pincer formed by a lineage-specific sequence insertion βi4, β-lobe and σ70 (Fig. 3b–e).

The C-terminal globular domain of CedA (residues 27–80), structurally similar to dsDNA-binding proteins[16], does not contact DNA. Instead, it binds βi4 via predominantly hydrophobic contacts (Fig. 3d). An N-terminal part of CedA, which was reported as unstructured[8,16], stretches along the DNA-facing side of β-lobe and the very N terminus (residues 5–10), bridging the gap between the β-lobe and σ70 region 1.2 (Fig. 3c,e), effectively locking the melted DNA in the RNAP main channel.

The N terminus of CedA forms several hydrogen bonds with σ70 (Fig. 3d). The only side-chain-specific contact was found between CedA Asn9 and σ70 Arg385, whereas other contacts involve main-chain atoms (Fig. 3e). A sequence alignment of *E. coli* σ factors (Extended Data Fig. 3) reveals a high degree of similarity between σ70 and σS in region 1.2, while σ32 appears more distant, and other σ factors have no apparent similarity. A structural alignment (Extended Data Fig. 4) mirrors the sequence comparison, implying that CedA could recognize RNAP holoenzyme bound to σS, although strength of the interaction would be diminished due to the loss of a contact between CedA Asn9 and σ70 Glu109 replaced by a proline in σS and σ32 (Extended Data Fig. 4).

A close spatial proximity of the CedA N terminus to σ70 regions 1.1 and 2, which are bound to a nontemplate DNA strand, explains why an apparent DNA binding site of CedA was identified as a −10 σ70 promoter consensus sequence. In spite of the lack of any direct contacts between CedA and DNA, formaldehyde crosslinking would tie CedA to the σ70-bound −10-consensus DNA.

A closer comparison of our structure with other models of the initiation complex (IC) revealed a major difference with DksA/TraR-bound structures[22,23]. DksA, and its homolog TraR, bind in the secondary channel of RNAP affecting kinetics of transcription initiation in a promoter sequence-dependent manner[24,25]. On binding to the IC, these proteins cause a major conformational change involving the movement of β-lobe/βi4 to form contacts with DksA (or TraR) (Fig. 4a), which widens the main channel of RNAP, thus facilitating a reversal of the open-promoter complex (RPo)[22,23]. This conformational change appears to be incompatible with our structure, as CedA would prevent such movement by keeping β-lobe/βi4 tied to σ70 (Fig. 3c). This observation predicts that CedA has an overall stabilizing effect on RPo by keeping DNA melted in the major groove of RNAP and, thus, also counteracting the effect of DksA (or TraR).

## CedA stabilizes an open-promoter complex, counteracts DksA

To test our structure-based predictions, we performed in vitro transcription reactions using DNA templates containing DksA-sensitive and -resistant promoters of the *E. coli* ribosomal operon *rrnB*. To investigate a potential effect of CedA on transcription initiation, the reactions were performed with a limited set of nucleotides (ATP and CTP only) allowing the synthesis of short RNA products before the transition to a productive elongation phase.

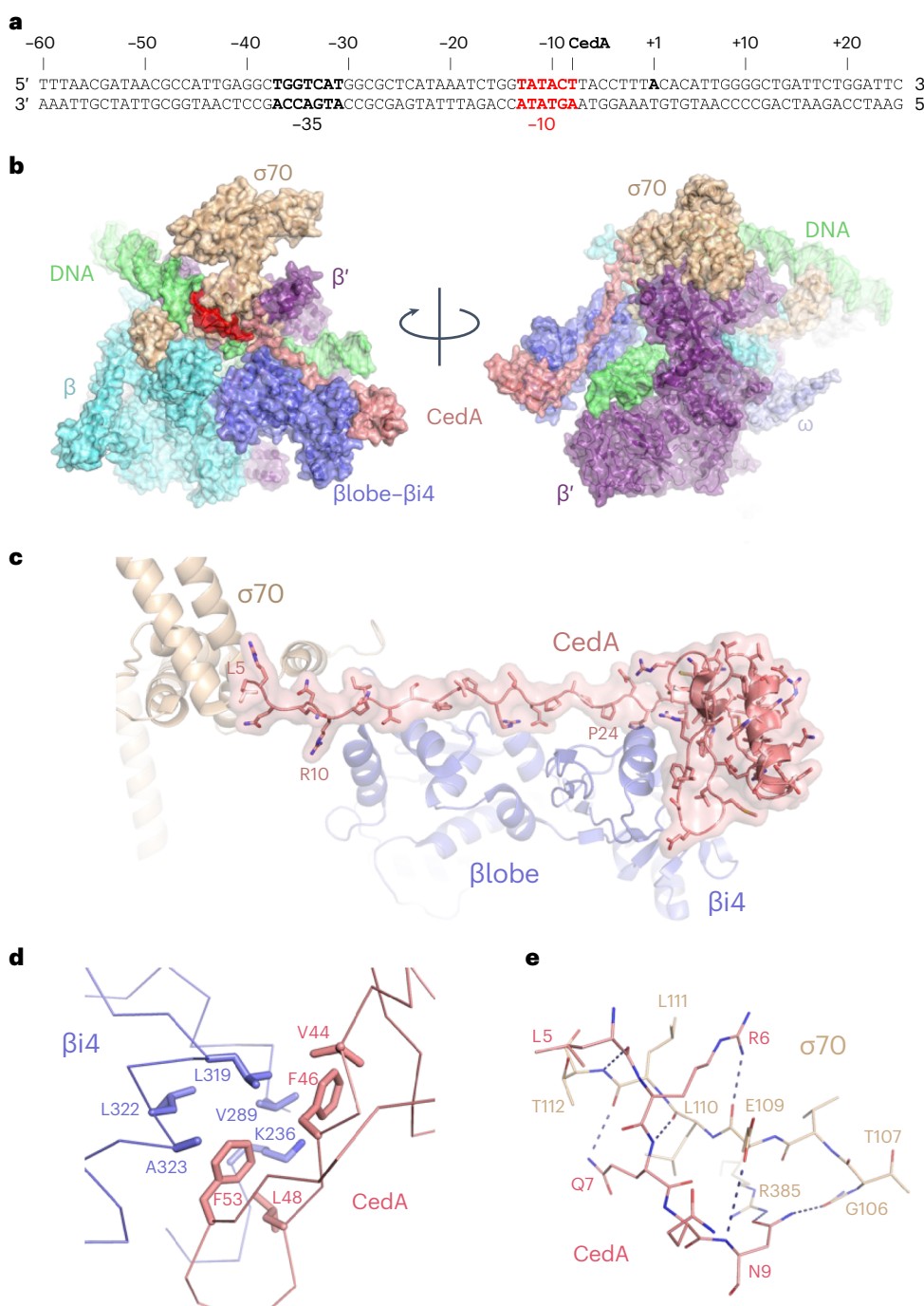

**Fig. 3 | Cryo-EM structure of the CedA-bound open-promoter complex.**
**a**, DNA sequence of ssrAp promoter. Position of CedA peak is marked, and the −10-consensus sequence is shown in red. **b**, Surface representation of CedA-bound open-promoter complex shown from two angles. Colors represent individual subunits and their parts: DNA (green), β subunit (cyan), βlobe/βi4 (dark blue), β′ subunit (purple), ω subunit (light blue), σ70 (wheat) and CedA (salmon). **c**, Close-up view of CedA interaction with βlobe/βi4 and σ70. **d**, Interface between CedA C-terminal domain and βi4 formed by hydrophobic residues. **e**, Hydrogen bonds formed between the N terminus of CedA and σ70 region 1.2.

The transcriptional assay in the presence of DksA and its coeffector ppGpp confirmed the inhibition of DksA-sensitive promoter *rrnB*P1, and no effect on *rrnB*P2 (Fig. 4b). When CedA was present, DksA–ppGpp were no longer able to inhibit *rrnB*P1 transcription, whereas *rrnB*P2 transcription was mildly suppressed by CedA independently of DksA–ppGpp. These results validate our prediction of CedA counteracting DksA, and also showed that CedA may have its own IC inhibitory activity. This inhibitory effect of CedA may result from an increased lifetime

of RPo, leading to a delayed promoter clearance and, as a result, a reduced reaction turnover rate.

Next, we compared the effect of CedA on the RPo lifetime. Following the RPo assembly, transcription reactions were incubated with heparin, a nonspecific competitor of DNA that prevents re-initiation of transcription. After various incubation times with heparin, the reactions were chased with ribonucleoside-5′-triphosphates (NTPs) to allow for a single-round RNA synthesis (Fig. 4c). CedA affected both

**Table 1 | Statistics for data collection and model refinement of cryo-EM determination**

| | CedA–RNAP complex (EMDB-29423), (PDB 8FTD) |
|---|---|
| **Data collection and processing** | |
| Magnification | 105,000 |
| Voltage (kV) | 300 |
| Detector | Gatan K3 |
| Total images collected | 3,596 |
| Electron exposure (e⁻/Å²) | 60 |
| Defocus range (μm) | 1.7–2.5 |
| Pixel size (Å) | 0.852 |
| Symmetry imposed | $C1$ |
| Initial particle images (no.) | 540,000 |
| Final particle images (no.) | 177,841 |
| Map resolution (Å) | 2.76 |
| FSC threshold | 0.143 |
| **Refinement** | |
| Initial models used (PDB codes) | 6OUL, 2BN8 |
| Model-to-map fit | 0.8454 |
| R.m.s. deviations | |
| Bond lengths (Å) | 0.013 |
| Bond angles (°) | 1.449 |
| **Validation** | |
| MolProbity score | 1.89 |
| Clashscore | 4.62 |
| Poor rotamers (%) | 0.03 |
| Ramachandran plot | |
| Favored (%) | 95.13 |
| Allowed (%) | 4.87 |
| Disallowed (%) | 0.00 |

promoters in a positive way resulting in the increased amount of synthesized RNA after the incubation with heparin. This result shows that CedA does indeed increase the lifetime of RNAP-promoter complex by preventing it from dissociation, hence the increased resistance to heparin.

Finally, to verify that the observed effects were due to constrains imposed by CedA on RNAP plasticity at the promoter, we examined a CedA mutant (CedAΔ11N) lacking the first 11 amino acids. Based on the structure, this truncated mutant should not bind σ⁷⁰ and, hence, cannot prevent β-lobe–βi4 from the DksA-induced conformational change. As predicted, CedAΔ11N failed to counteract DksA-mediated inhibition of transcription at *rrnB*P1 (Fig. 4d).

## CedA enhances tolerance to oxidative stress and certain antibiotics

CedA is encoded by a single-gene operon. Its promoter is directly adjacent to *katE* promoter, facing the opposite direction (Fig. 5a). KatE (HPII) is one of the two catalases in *E. coli*, the expression of which depends on general stress sigma factor σ^S [26]. The direct proximity between *katE* and *cedA* promoters suggests the two genes may share a common function. Indeed, the expression of *katE* and *cedA* is induced approximately sixfold by peroxide (Fig. 5b). Furthermore, cells overexpressing CedA became more resistant to peroxide than control cells (Fig. 5d).

As many antibiotics promote oxidative stress[27], we also examined whether CedA offers any protection against different classes of antibiotics. We found that cells overexpressing CedA become more resistant to aminoglycosides gentamycin and kanamycin (Fig. 5d). CedA appears to have no effect on cellular tolerance to quinolones (ciprofloxacin) and macrolides (erythromycin) (Fig. 5d). Concordantly, aminoglycosides, but not erythromycin or ciprofloxacin, also stimulated *cedA* transcription (Fig. 5c).

The colony forming ability of Δ*cedA* cells was greatly diminished in the presence of RNAP inhibitor rifampicin (Fig. 6a). A plasmid for overexpression of full-length CedA, but not its truncated version, not only abolished rifampicin sensitivity of Δ*cedA* cells, but also allowed them to grow at higher concentrations of rifampicin comparing to the wild-type (WT), highlighting CedA involvement in rifampicin tolerance as well.

To identify whether rifampicin tolerance resulted from a direct modulation of RNAP activity by CedA, we performed transcription reactions in the presence of rifampicin using a DNA template containing a generic σ⁷⁰ promoter T7A1. CedA activated transcription resulting in a higher overall yield of RNA; however, the degree of RNAP inhibition by rifampicin was not affected (Fig. 6b). This result indicates that CedA does not alter the mode of interaction between RNAP and rifampicin, but contributes to antibiotic tolerance by promoting residual transcription in the presence of the drug.

## CedA modulates hundreds of *E. coli* genes

As antibiotic tolerance may involve multiple differentially expressed genes responding to CedA, we performed an RNA sequencing (RNA-seq) experiment to compare gene expression between WT bacteria and strains lacking or overexpressing CedA.

The WT, Δ*cedA* and Δ*cedA* cells transformed with pCedA were grown in Luria-Bertani (LB) medium at 37 °C to an optical density (OD$_{600}$) of roughly 0.15–0.2, before cultures were supplemented with rifampicin (10 μM). After 1 hour of incubation, bacteria were harvested and used for RNA isolation. Differentially expressed genes responding to both CedA and rifampicin were identified by factorial analysis of variance (ANOVA) test using log$_2$-transformed normalized read counts as a measure of messenger RNA (mRNA) levels.

We identified 1,309 genes ($q < 0.01$) responding to CedA and rifampicin (Supplementary Table 5). The overall pattern of gene expression (Fig. 6c) shows that in both control and rifampicin conditions, the overexpression of CedA has a much larger overall effect, whereas the pattern of gene expression in Δ*cedA* mostly follows that of the WT, with many genes yet significantly affected. To further clarify the effect of CedA, we filtered differentially expressed genes based on the magnitude of the response, keeping only those with log$_2$ fold change above 0.5 when compared to the WT. Both up- and down-regulated genes represented a significant fraction in both strains (Fig. 6d), with a larger number of genes alternatively regulated in a strain overexpressing CedA.

Many genes affected by CedA were transcription factors, RNA modifying enzymes and ribonucleases (Supplementary Table 5), suggesting that globally altered mRNA levels could be a result of an indirect action of CedA. Thus, to identify direct effects of the protein, our next step was to look at the effect of CedA on genes that promoters were associated with it based on ChIP–seq (Supplementary Table 4). All the ten promoters that passed the filtering criteria ($q < 0.01$, log$_2$ fold change above 0.5) were downregulated in rifampicin-treated Δ*cedA* cells (Fig. 6e). Overexpression of CedA effectively reversed the inhibitory effect of rifampicin for eight of them, bringing the expression level of downstream genes to that of the WT or above.

Results of RNA-seq experiment show that CedA modulates gene expression in condition-dependent and promoter-dependent manner. An apparent transcriptional repression in cells grown without the antibiotic may result from the increased lifetime of RPo leading

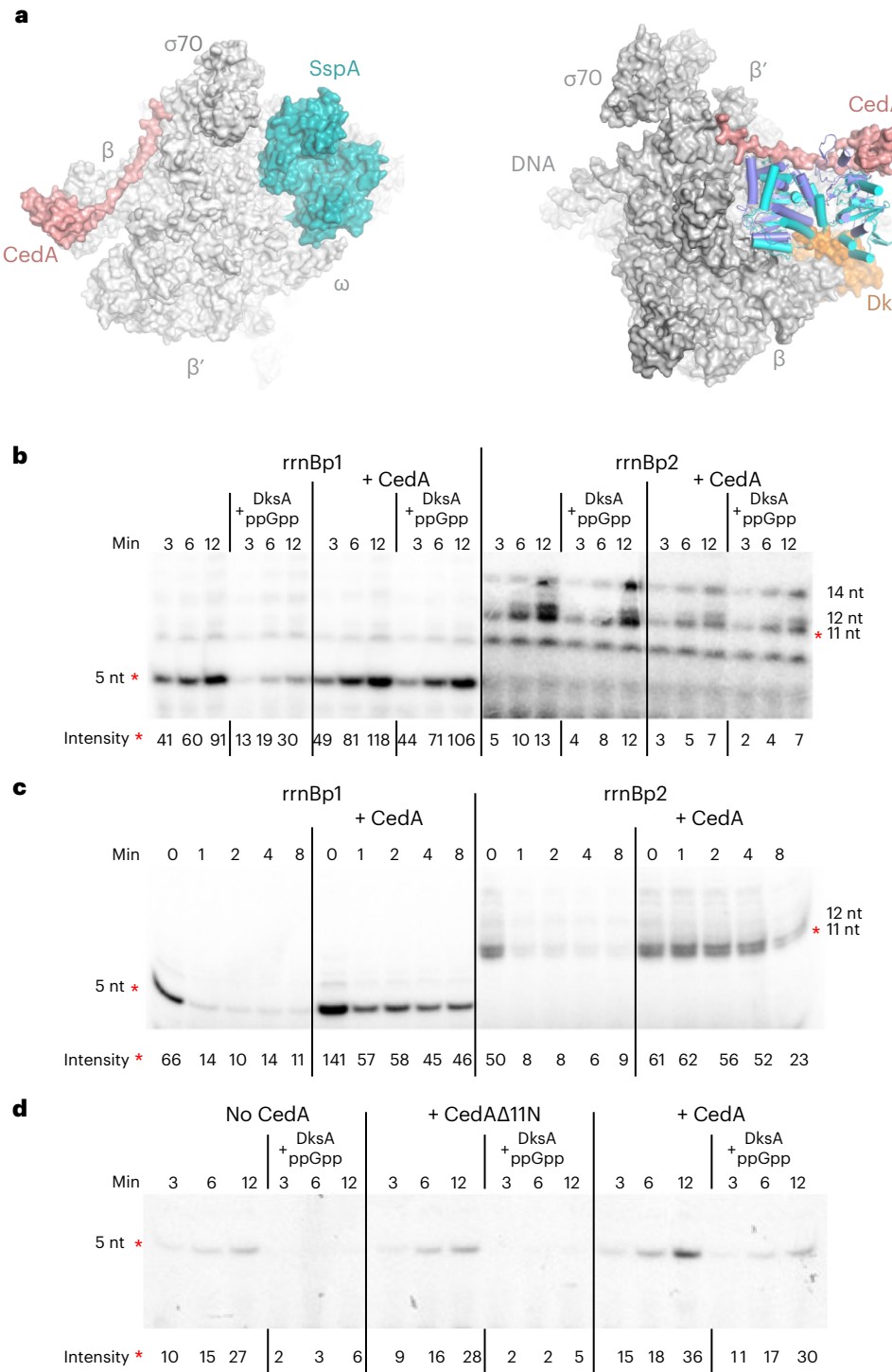

**Fig. 4 | Modulation of transcription initiation by CedA. a,** Structural comparison of CedA-, SspA- and DksA-bound open-promoter complex. Structures of SspA- (PDB 7C97)[28] (left) and DksA-bound (PDB 7KHI)[23] (right) RPo were aligned with CedA-bound structure using PyMol v.2.4. SspA and DksA are shown in teal and orange, respectively. Conformational change in β-lobe/βi4 on DksA binding is shown by an arrow. The structure of β-lobe–βi4 in the DksA-bound complex is shown in cyan, the CedA-bound complex is dark blue. **b–d,** Representative autoradiograms of the RNA products from the transcription initiation assay using *rrnB*P1 or *rrnB*P2 promoters. Relative intensity values for the bands marked with red asterisks were obtained using ImageQuant software

(Methods). Normalization between independent experiments was achieved by using the bands with highest intensity readings. Standard deviation (s.d.) was calculated from at least three independent experiments. **b,** Reactions were incubated for 3, 6 and 12 min in the presence of DksA–ppGpp and/or CedA. **c,** After the assembly of the open-promoter complex, reactions were either immediately supplemented with NTPs (time point 0) or mixed with heparin and incubated for 1, 2, 4 and 8 min before the addition of NTPs. **d,** Reactions initiated from *rrnB*P1 were incubated for 3, 6 and 12 min with or without DksA–ppGpp together with CedA or its truncated version lacking the 11 N-terminal amino acids (CedAΔ11N).

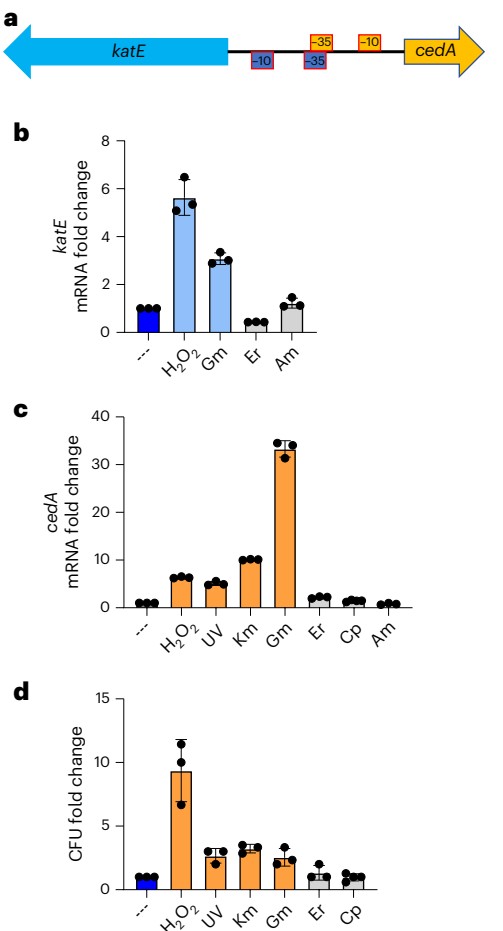

Our experimental setup relies on in vivo crosslinking with formaldehyde, a fast acting and robust crosslinking agent, to fix protein–protein complexes in their natural environment before affinity isolation and analysis by MS. A typical experiment would identify hundreds of proteins, among which only 5–10% top ranking may be considered as true interactors. Proteins that bind to affinity media nonspecifically were excluded from the analysis as false positives. In addition to ranking proteins by their abundance, we measured degree of specificity and enrichment, which all together were summarized in a combined rank, greatly improving the identification of RNAP-interacting partners. With the same analysis applied to CedA, we were able to confidently call RNAPσ70 holoenzyme as the primary in vivo interacting partner of CedA. A complementary ChIP–seq analysis identified σ70 promoters as a target for CedA in *E. coli* genome. Overall, a combination of high-throughput methods, as exemplified in this study, provides an efficient way to determine interacting macromolecules and infer their function.

To understand the mechanism of CedA we used cryo-EM to solve the structure of CedA bound to the RPo. With the high-resolution map, we were able to trace most of the CedA, which acquires ladle-shaped conformation when bound to RNAP. To our surprise, CedA, which was predicted to bind DNA via its C-terminal domain[8], interacts exclusively with RNAP β subunit and σ70, and not with DNA. A cup of the ladle-shaped CedA molecule, the C-terminal globular domain, binds at the tip of β-pincer that is formed by βi4 of RNAP. Specificity of CedA for a lineage-specific sequence insertion explains why it is conserved in proteobacteria. The rest of the molecule, the elongated N-terminal part of CedA (handle of the ladle) stretches along β-lobe and binds to σ70 region 1.2. Sequence alignment of *E. coli* σ factors suggests that CedA may, in principle, also bind σS and σH holoenzymes. However, our proteomic analysis of CedA interactome does not support alternative σ as plausible interactors in vivo.

Based on our structural analysis, we predicted that CedA should have an overall stabilizing effect on RPo by restricting the movement of β-lobe, while keeping it in a closed conformation, contrasting an open conformation induced by DksA–ppGpp or TraR[22,23]. In vitro experiments confirmed our predictions. We found that CedA stimulates transcription from DksA-sensitive promoter *rrnB*P1 and has a stabilizing effect that increased the lifetime of RPo when challenged with heparin.

CedA renders *E. coli* more resistant to oxidative stress and antibiotics that inhibit translation (aminoglycosides) and transcription (rifampicin). Moreover, *cedA* is strongly induced in response to aminoglycosides and peroxide, demonstrating the involvement of CedA in bacterial tolerance. When tested in vitro, CedA activated residual transcription in the presence of rifampicin, but did not change RNAP sensitivity to the antibiotic.

RNA-seq analysis shows that CedA affects hundreds of genes under normal growth conditions. We identified more than 1,000 genes that were affected by rifampicin treatment in CedA-dependent manner. The list is vast, which precludes us from pinpointing any specific candidate, or their combination, underlying rifampicin tolerance. However, we analyzed the expression of genes located immediately downstream of CedA-associated promoters identified by ChIP–seq to show that in cells overexpressing CedA these promoters were upregulated by rifampicin treatment, whereas in the deletion strain they were suppressed. Thus, we conclude that the stimulation of transcription by CedA contributes to rifampicin tolerance. A similar global transcriptional response mediated by CedA is likely to underlie tolerance to other clinically relevant antibiotics we examined here and to oxidative stress, highlighting CedA as a new promising antimicrobial target, especially in light of its positive retention in uropathogenic *E. coli*[11].

**Fig. 5 | CedA renders *E. coli* cells more tolerant to peroxide and aminoglycosides. a**, Schematics of cedA promoter region. Arrows indicate the direction of transcription. The −10 and −35 elements of *katA* and *cedA* promoters are colored in blue and yellow, respectively. **b,c**, katA (**b**) and cedA (**c**) mRNA changes in response to different stress conditions as determined by RT–qPCR. The amounts of katA and cedA mRNA were normalized to that of gapA. The resulting changes in the reaction threshold cycle values (ΔCt) were compared to nonstressed control (dark blue bars). −log2 differences in Ct values (ΔΔCt) were plotted as fold changes. Significant changes ($P < 0.01$) are shown as blue (**b**) and yellow (**c**) bars for *katE* and *cedA*, respectively, nonsignificant changes are shown with gray bars. H2O2 is 2 mM, UV is 64 J m−2, Km is kanamycin (50 μg ml−1), Gm is gentamycin (20 μg ml−1), Er is erythromycin (50 μg ml−1), Cp is ciprofloxacin (1 μg ml−1) and Am is ampicillin (100 μg ml−1). **d**, Survival rate of *E. coli* cells treated with the stressors as in **c** following the plasmid-born cedA gene induction. Significant changes in CFUs ($P < 0.05$) are shown as yellow bars, nonsignificant changes are gray bars. Values (fold change) were normalized to an empty vector control and compared to a nonstressed condition (dark blue bar). Datasets are presented as mean values ± s.d. *P* values were calculated by using unpaired *t*-test with Welch's correction. Each qPCR experiment was performed at least three times with independently produced mRNA, survival tests were each performed in independent triplicates.

to delayed promoter clearance, as we proposed to explain the inhibitory effect of CedA on multi-round transcription directed by *rrnB*P2 (Fig. 4b). Stimulation of transcription by CedA appears to be crucial for rifampicin tolerance and potentially involves promoters that are particularly sensitive to the antibiotic.

## Discussion

In this study, we combined chemical crosslinking and affinity isolation followed by MS-based proteomics and NGS to identify RNAP-associated protein CedA as a global transcription initiation factor.

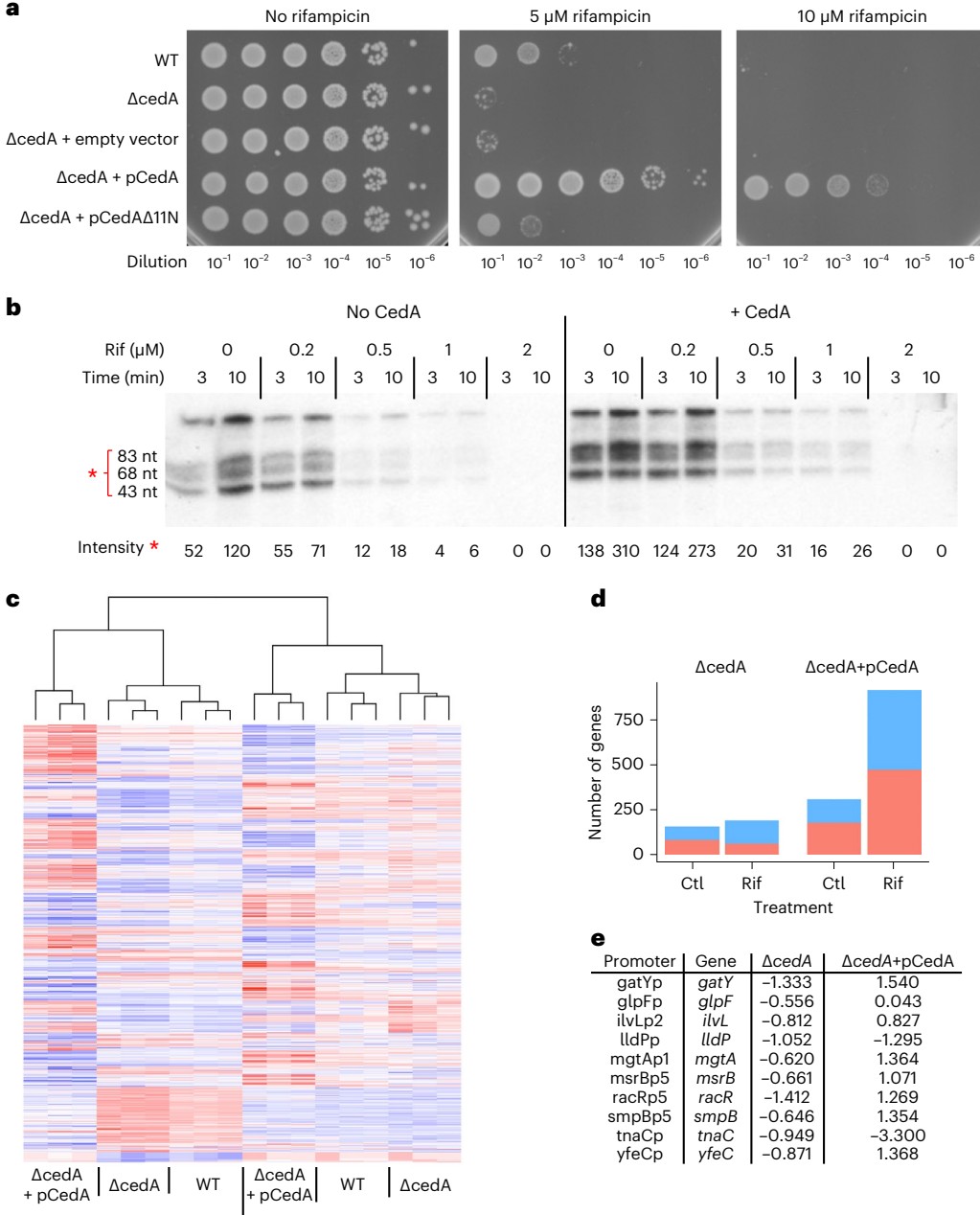

**Fig. 6 | CedA promotes rifampicin tolerance and alters global gene expression in E. coli. a**, Representative efficiencies of colony formation of parent WT, ΔcedA or ΔcedA mutant transformed with plasmids bearing no insert (empty vector), full-length cedA (pCedA) or truncated cedA lacking the 5′-terminal 11 codons (pCedAΔ11N). Bacteria were grown overnight on LB agar plates containing 10 ng ml⁻¹ anhydrotetracycline (inducer) and 0, 5, 10 μM rifampicin. **b**, Autoradiograms of the RNA products of multi-round transcription reactions performed in the presence of rifampicin and CedA using DNA template containing T7A1 promoter. Relative intensity values and s.d. for the bands marked with an asterisk were calculated as in Fig. 4 from three independent experiments. **c**–**e**, Differentially expressed E. coli genes regulated by CedA. **c**, Heatmap representation of a change in mRNA levels of differentially expressed genes (factorial ANOVA q < 0.01) in response to the cedA deletion or overexpression and treatment with rifampicin. Upregulated genes are shown in red, downregulated are in blue. **d**, Number of up- (red) and downregulated (blue) genes filtered by q value below 0.01 and log₂ fold change above 0.5. The log₂ fold change was calculated for the deletion (ΔcedA) and overexpression (ΔcedA+pCedA) strains relative to the WT. **e**, List of promoters associated with CedA that responded to rifampicin treatment (q < 0.01, log₂ fold change >0.5) in ΔcedA strain. Columns ΔcedA and ΔcedA+pCedA show magnitude of change of gene expression levels (log₂ fold change) relative to the WT in the presence of rifampicin in cedA deletion and overexpression strains, respectively.

## Online content

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

## Methods

### Bacterial strains and plasmids

*E. coli* MG1655 WT and its derivatives were used in ChIP–seq and RNA-seq experiments, analysis of protein interactions in vivo, and rifampicin resistance. *E. coli* MG1655 *rpoC*-3× FLAG was among the strains in laboratory collection[29], *E. coli* MG1655 *cedA*-3× FLAG and *E. coli* MG1655 Δ*cedA* were prepared in this study. *E. coli* BL21(DE3) and NEB Turbo Competent *E. coli* (NEB) were used for proteins overexpression and DNA cloning, respectively.

paTc-based vectors[29] were used for tetracycline-inducible expression of CedA (paTc-*cedA*) and its truncated variant lacking first 11 amino acids (paTc-*cedA*Δ11N). Plasmids pVS10 [30] and pSumoH10-*cedA* were used for the overexpression of *E. coli* RNAP and CedA, respectively.

### Chromosomal deletions and tagging

The modification and/or deletion of chromosomal genes were done using λ Red recombinase system as described elsewhere[31]. To incorporate a 3× FLAG epitope at the C terminus of a target protein, a kanamycin resistance cassette from pKD4 was first PCR-amplified using a pair of primers (Supplementary Table 6) introducing a 3× FLAG-coding region at the 5′ terminus of the cassette. Then a 3× FLAG-bearing cassette was amplified with a gene-specific pair of primers to introduce a 3× FLAG-encoding sequence at the 3′ end of a target gene immediately before the stop codon. For a gene deletion, a kanamycin resistance cassette was amplified with a pair of primers including the sequences flanking an open reading frame of a target gene. Resulting DNA fragments were introduced by electroporation into *E. coli* MG1655 bearing pKD46. After 2 h of recovery in SOC media at 30 °C, bacteria were plated on LB agar containing 50 μg ml⁻¹ kanamycin and colonies were grown at 37 °C for 16–20 h. Individual colonies were restreaked on LB plates containing 50 μg ml⁻¹ kanamycin, grown overnight at 37 °C and screened for loss of pKD46, which was confirmed by lack of ampicillin resistance when bacteria were grown on LB agar containing 100 μg ml⁻¹ carbenicillin.

DNA from positive clones was extracted using a Monarch Genomic DNA Purification kit (NEB). DNA insertion was confirmed by amplification of a fragment including both genomic and kanamycin resistance cassette sequences followed by the sequencing of amplified DNA (Macrogen).

### Plasmid construction

Plasmids were constructed using a NEBuilder HiFi DNA Assembly kit (NEB) according to the manufacturer's instructions. Assembly reactions contained reverse-amplified vector DNA (paTc or pSumoH10) treated with Dpn I and a DNA fragment comprising the *cedA* gene amplified from *E. coli* MG1655 chromosome flanked by vector-specific sequences. After transformation, NEB Turbo *E. coli* were selected on LB agar containing appropriate antibiotic (100 μg ml⁻¹ carbenicillin for paTc, 50 μg ml⁻¹ kanamycin for pSumoH10). Individual clones were then grown in liquid LB, plasmid DNA was isolated using PureLink Plasmi Miniprep kit (Thermo) and correct inserts were confirmed by sequencing (Macrogen).

### In vivo crosslinking

*E. coli* MG1655 *rpoC*-3× FLAG or *cedA*-3× FLAG strains were grown overnight at 37 °C in LB media containing 50 μg ml⁻¹ kanamycin. Then 0.3 ml of overnight cultures were inoculated into 30 ml of fresh LB contained in 125-ml flasks and new cultures were grown to mid-log phase (OD₆₀₀ 0.4–0.6) at 37 °C with shaking at 250 rpm. For crosslinking, 16% methanol-free formaldehyde (Fisher) was added to 0.3% final concentration and the incubation continued for 10 min. To neutralize formaldehyde, 2 M glycine was added to cultures at 0.2 M final concentration for 5 min. Bacteria were then collected by 10 min of centrifugation at 4,000*g*, 4 °C. Pellets were resuspended in 0.3 ml of lysis buffer: 50 mM Tris-HCl pH 7.5, 150 mM NaCl, 10 mM MgCl₂, 10% glycerol, 0.5%

Triton X-100 and protease inhibitors cocktail (Roche) and stored at −20 °C. Lysis buffer for ChIP–seq experiments contained 1 mM EDTA instead of 10 mM MgCl₂.

### Immunoprecipitation

Bacterial samples that were stored frozen as suspension in a lysis buffer were thawed on ice. Lysozyme (Roche) and Pierce Universal Nuclease (Thermo) were both added to 0.5 U μl⁻¹ final concentration. To prevent DNA degradation in the ChIP samples, Pierce Universal Nuclease was replaced with 5 μg ml⁻¹ RNase A. Bacteria were then disrupted by 20 cycles of sonication in Bioruptor Pico (Diagenode) with each cycle lasting for 30 s between 30 s breaks. In ChIP samples, the sonication ensured DNA fragmentation to an average size 150 bp. Lysates were clarified by 10 min of centrifugation at 20,000*g* at 4 °C and transferred to the new tubes.

To capture FLAG-tagged proteins, Pierce Anti-DYKDDDDK Magnetic Agarose (Thermo) pre-equilibrated in the lysis buffer was added at 5 μl of packed beads per sample followed by 2 h of incubation at 4 °C on a rotary mixer. Beads were washed four times with 1 ml of ice-cold wash buffer (20 mM Tris-HCl pH 7.5, 500 mM NaCl, 1 mM EDTA, 0.1% Triton X-100) and kept on ice before proceeding to the next step of either on-beads digestion of proteins with trypsin for liquid chromatography with MS (LC–MS) analysis, or DNA extraction for the construction of sequencing libraries.

### On-beads proteins digestion

Following the immunoprecipitation procedure, beads were washed twice with 1 ml of ice-cold NH₄HCO₃ to exchange buffer and remove detergent as described[17]. Washed beads were resuspended in 50 μl 50 mM NH₄HCO₃ containing 20 ng μl⁻¹ SOLu trypsin (Sigma) and incubated at 25 °C with vigorous shaking overnight. Beads were then pelleted and supernatants were transferred to the new tubes.

### In-solution protein digestion

To identify a protein composition in *E. coli* lysates, proteins from 5 μl of clarified lysates were precipitated with acetone. Four volumes of acetone chilled to −20 °C were added to clarified lysates followed by 1 h of incubation at −20 °C. Precipitates were collected by 10 min of centrifugation at 20,000*g* at 4 °C. Pellets were rinsed with 80% acetone and let dry in air for 15 min. Dried pellets were dissolved in 5 μl of 50 mM NH₄HCO₃, 8 M urea, then mixed with 50 μl of 50 mM NH₄HCO₃ containing 20 ng μl⁻¹ SOLu trypsin (Sigma) and incubated overnight at 25 °C.

After the overnight incubation, digestion reactions were mixed with an equal volume 2% HFBA, incubated at room temperature for 5 min and clarified by 5 min of centrifugation at 16,000*g*. Peptides from supernatants were then desalted using Pierce C18 spin tips (Thermo) according to the manufacturer's instructions and dried under a vacuum. Dried peptides were reconstituted in 0.1% formic acid and concentration was measured at 205 nm on a Nanodrop One (Thermo).

### LC–MS analysis of peptides

Depending on concentration, 0.5–2 μg of peptides were analyzed using Orbitrap Fusion Lumos mass spectrometer coupled with Dionex Ultimate 3000 RSLC Nano UHPLC (Thermo). After capturing on 2 cm long (0.2 mm internal diameter, 5 μm particle size) Acclaim PepMap 100 C18 trap column (Thermo) and washed with buffer A (0.1% formic acid) for 5 min at 5 μl min⁻¹, peptides were resolved on a 50 cm long (75 μm ID, 2 μm particle size) EASY-Spray column (Thermo) over a 80 min long gradient 2 to 32% buffer B (100% acetonitrile, 0.1% formic acid) at a flow rate of 0.2 μl min⁻¹, followed by a steep increase to 90% buffer B for 5 min and 5 min of step elution with 90% buffer B. Data collection was controlled with an XCALIBUR v.4.1 (Thermo). The data-dependent acquisition method was based on a published protocol[32] except that each cycle was set to last for 2 s instead of 3 s.

## MS data analysis

Raw MS data were processed using a MaxQuant v.2.0.1 software package[33]. Protein database included *E. coli* MG1655 proteome (https://www.uniprot.org/proteomes/UP000000625) and a list of common protein contaminants. The search engine was run with default parameters for mass tolerance, up to two missed trypsin cleavages were allowed, variable modifications were methionine oxidation, acetylation of protein N terminus and cysteine carbamidomethylation. 'Second peptide' and 'Match between runs' options were enabled. Label-free quantitation was used for MS1-based peptides quantitation.

Further analysis was done in R using quantitation data from a MaxQuant output recorder in 'proteinGroups.txt' file. Protein abundance was normalized to an overall abundance and a protein molecular weight, and $\log_2$-transformed. Missing intensities recorded as zero were converted to explicit missing values. Proteins were then ranked by their abundance in a target pulldown, by a difference in the abundance between target and nonspecific pulldowns, and by a difference in the abundance between a target pulldown and unfractionated lysate. Next, overall rank ranging from zero to one was calculated as a sum of three ranks. Highly ranking proteins (close to 1) identified in each of three experimental replicates were considered as part of a protein complex involving bait protein–RpoC or CedA.

## Extraction of DNA and preparation of ChIP–seq libraries

Following the immunoprecipitation, washed beads were resuspended in DNA elution buffer containing: 50 mM Tris-HCl pH 8.8, 100 mM NaCl, 10 mM EDTA, 1% SDS and 10 U ml$^{-1}$ Proteinase K (NEB), followed by 18 h of incubation at 56 °C with vigorous shaking. Beads were then pelleted and supernatants were transferred to the new tubes. DNA containing in supernatants was cleaned up using Invitrogen PureLink PCR Purification kit (Thermo). Purified DNA was stored at −20 °C.

The NEB Next Ultra II DNA Library Prep kit for Illumina (NEB) was used to prepare DNA libraries according to the manufacturer's instructions. Libraries were amplified using primers supplied with an NEB Next Multiplex Oligos for Illumina kit, Index Primers Set 3 (NEB). Amplified libraries were cleaned up using Sample Purification Beads (NEB) included in the kit. DNA concentration was measured on Qubit (Thermo) and quality analysis was done on a TapeStation (Agilent).

## ChIP–seq data analysis

Here, 75-bp paired reads were aligned to *E. coli* MG1655 genome (National Center for Biotechnology Information Reference Sequence NC_000913.3) using BOWTIE2 (http://bowtie-bio.sourceforge.net/bowtie2/index.shtml, v.2.4). For each genome position, sequencing depth was counted using the SAMTOOLS DEPTH tool (https://www.htslib.org, v.1.13). Resulting values were $\log_2$-transformed and smoothed using rolling average over a 50 bp wide window. After subtracting background (values obtained from nonspecific pulldown) and baseline values (mode of values across all genomic positions), ChIP–seq peaks were called for the positions where local maxima across 50 bp windows had an intensity at least 3σ above zero.

## Analysis of differential genes expression using RNA-seq

Bacterial cultures of *E. coli* MG1655 WT, Δ*cedA*::kan and Δ*cedA*::kan bearing paTc-*cedA* plasmid were grown overnight in LB media at 37 °C, 250 rpm. Media was supplemented with 50 μg ml$^{-1}$ kanamycin for Δ*cedA*::kan strain, 50 μg ml$^{-1}$ kanamycin and 100 μg ml$^{-1}$ carbenicillin for Δ*cedA*::kan/paTc-*cedA* strain. Overnight cultures were inoculated at 1/100 vol. into fresh LB containing 10 ng ml$^{-1}$ anhydrotetracycline and grown for 2 h at 37 °C, 250 rpm until the early log phase. From each culture, two 0.5-ml aliquots were taken: control aliquots were mixed with 5 μl of $H_2O$ mQ, and treated with 5 μl of 1 mM rifampicin (diluted with $H_2O$ mQ from 10 mM stock solution in dimethylsulfoxide). After an extra 1 h of incubation at 37 °C, 50 μl of 10% phenol in ethanol was added to prevent RNA degradation, and bacteria were pelleted by

centrifugation at 12,000*g*, 4 °C for 1 min. Media was aspirated and pellets were stored at −20 °C.

Frozen pellets were thawed on ice and resuspended in 50 μl of lysis solution: 50 mM Tris-HCl pH 8.0, 1 mM EDTA, 0.1% Triton X-100, 30 U μl$^{-1}$ rLysozyme (Roche), followed by 5 min incubation on ice. Samples were then mixed with 350 μl of 1× RNA protection reagent and RNA was purified using a Monarch total RNA extraction kit (NEB) according to the manufacturer's instructions. Purified RNA was stored at −80 °C.

RNA-seq libraries were prepared with NEB Next Ultra II Directional RNA-seq kit for Illumina (NEB) after ribosomal RNA (rRNA) depletion (NEB Next rRNA depletion kit by NEB) using 1 μg of total RNA. RNA-seq libraries were amplified and analyzed as described for ChIP–seq libraries.

## RNA-seq data analysis

The 75-bp paired reads were aligned to *E. coli* MG1655 genome using BOWTIE2. Reads mapping to annotated regions were counted in strand-specific manner using a featureCounts function of Rsubread R package[34], including only those reads with the mapping quality above ten and the length overlap above 0.5. Read counts were then normalized to gene length, transformed to $\log_2$ scale and variance stabilizing normalization was applied[35]. To identify differentially expressed genes, factorial ANOVA test was performed followed by multiple testing correction[36,37]. Genes were called differentially expressed if the associated *q* value was below 0.01.

## NGS

The libraries were sequenced on an Illumina NextSeq 500 instrument in 2 × 75 bp paired end mode. For ChIP–seq experiments the sequencing depth was typically 10–20 M reads per sample. RNA-seq libraries were sequenced the depth of 25–30 M reads per sample.

## Proteins expression and purification

WT *E. coli* RNAP and σ[70] used in the work were purified as previously described[38]. WT *E. coli* DksA was purified according to[39]. The open reading frame of the *E. coli* cedA was cloned into the pSUMO vector, which has ten tandem Histidine tag (10× His) followed by SUMO tag. The plasmid was transformed into *E. coli* strain BL21(DE3) for overexpression and recombinant protein expression was induced with 0.5 mM isopropyl-β-D-thiogalactoside when the $OD_{600}$ reached 0.6. After 3 h at 37 °C, cells were collected for protein purification. Cell pellets were resuspended in lysis buffer (50 mM HEPES, pH 7.0, 500 mM NaCl, 5% (v/v) glycerol, 15 mM imidazole, 5 mM β-mercaptothanol) supplemented with complete, EDTA-free protease inhibitor cocktail tablets (Roche Applied Science) and lysed using sonication on ice. The cell debris was removed by centrifugation (40,000*g* for 45 min at 4 °C). The supernatant was applied to a HisTrap column (Cytiva) equilibrated in lysis buffer. The column was washed using the lysis buffer until the ultraviolet light at 280 nm ($UV_{280}$) absorption reaches the baseline. Protein was eluted with HisTrap Buffer (50 mM HEPES, pH 7.0, 5% (v/v) glycerol, 5 mM β-mercaptoethanol, 250 mM NaCl, 300 mM imidazole). Fractions containing recombinant 10× His-SUMO-CedA were analyzed by SDS–PAGE and then were subjected to tag removal by Ulp1 in dialysis buffer (25 mM HEPES, pH 7.0, 5% (v/v) glycerol, 5 mM β-mercaptoethanol, 100 mM NaCl) at 4 °C overnight. Sample was then applied on a HiTrap SP HP cation exchange chromatography column (5 ml) equilibrated in SP-A Buffer (25 mM HEPES, pH 7.0, 5% (v/v) glycerol, 5 mM β-mercaptoethanol, 100 mM NaCl). The protein was eluted with a linear gradient of NaCl 100 mM to 1 M in 20 column volumes (100 ml). The peak fractions containing CedA were pooled, concentrated, and further purified by a Superdex 75 (10/300) size exclusion column (Cytiva) equilibrated with buffer S (30 mM HEPES, pH 8.0, 200 mM NaCl, 5 mM $MgCl_2$, 0.5 mM TCEP). The peak fractions from the Superdex 75 column were pooled, concentrated to 5 mg ml$^{-1}$ concentration, flash frozen in liquid nitrogen and stored at −80 °C.

## Nucleic-acid scaffold preparation

Synthetic DNA oligonucleotides containing *ssrA* promoter sequence (−65 to +20) were purchased from Integrated DNA Technologies. In brief, the nucleic acids were dissolved in RNase-free water (ThermoFisher Scientific) at 1 mM concentration. Template DNA and nontemplate DNA were mixed at 1:1 ratio, annealed by incubating at 98 °C for 5 min, 75 °C for 2 min, 45 °C for 5 min and then decreasing the temperature by 2 °C for 2 min until reaching 25 °C. The annealed template DNA–nontemplate DNA hybrid was stored at −20 °C before use.

## Complex assembly for cryo-EM

The RNAP holoenzyme (Eσ[70]) was formed by mixing purified RNAP and a twofold molar excess of σ[70] and incubating for 30 min at room temperature. Eσ[70] was purified on a Superose 6 Increase 10/300 GL column in gel-filtration buffer (30 mM HEPES, pH 8.0, 100 mM KCl, 5 mM MgCl$_2$, 0.5 mM TCEP). The eluted Eσ[70] was concentrated to roughly 5.0 mg ml$^{-1}$ (roughly 10 µM) by centrifugal filtration. Annealed *ssrA* promoter DNA was added (threefold molar excess over RNAP) and the sample was incubated for 30 min at room temperature followed by another 30 min at room temperature after addition of CedA (fivefold molar excess over RNAP). The whole complex was purified on Superose 6 Increase 10/300 GL column equilibrated in gel-filtration buffer. The peak fractions containing RNAP-CedA from the Superose 6 column were analyzed by SDS–PAGE, pooled, concentrated to roughly 4 mg ml$^{-1}$ for cryo-EM sample preparation.

## EM data acquisition

Cryo-grid preparation was performed using FEI Vitrobot mark IV operated at 10 °C and 100% humidity. Aliquots of 4 µl of a freshly purified CedA-RPo (roughly 4 mg ml$^{-1}$), in the presence of 8 mM CHAPSO[38], was applied to glow-discharged holey carbon grids (Quantifoil Au, R1.2/1.3, 300 mesh). The grids were blotted for 3 s and flash-plunged into liquid ethane, precooled in liquid nitrogen. The cryo-grids were examined using FEI Arctica operating at 200 KV equipped with K3 direct electron detector, and then good cryo-grids were loaded to FEI Titan Krios electron microscope. All the cryo-images were recorded on Gatan K3 camera operated in super-resolution mode. The magnification is ×105,000 corresponding to a final pixel size of 0.852 Å by binning 2 of the original micrographs. For each image stack, a total dose of about 60 electrons were equally fractioned into 42 frames with a total exposure time of 4.2 s. Defocus values ranged from 1.7 to 2.5 µm. In total, 3,596 micrographs were collected using Leginon[40].

## Image processing

For cryo-EM datasets, beam-induced motion correction was performed using the MotionCorr2 through all frames[41]. The contrast transfer function (CTF) parameters were estimated by CTFFIND4[42]. About 540,000 particles were picked from 3,596 micrographs using Gautomatch in a template-free mode. RELION[43] was used for 2D class average and 3D classification. Then 177,841 particles were imported to cryoSPARC[44] for final refinement. A reported 2.76 Å map was generated after homogenous refinement and local CTF refinement. All the visualization and evaluation of the map was manipulated within Chimera[45], and the local resolution map was calculated using ResMap[46].

## Model building and refinement

Protein Data Bank (PDB) 6OUL and 2BN8 were used as the template for model building, and *ssrA* promoter sequence was manually placed in COOT[47]. The resolution was high enough for us to accurately assign the residues of N-terminal of CedA. Phenix[48] was used for real space refinement. The structural and refinement statistics were summarized in the table (Table 1).

## In vitro transcription

DNA templates were constructed by PCR amplification and purified from agarose gel using Qiagen gel extraction kit according to the manufacturer.

The sequence of *rrnB*P1, *rrnB*P2 and A1 templates are shown below. The non-transcribed part is italicized. Primers used for PCR amplification are underlined. Bold A and C mark position of the start of transcription for P1 and P2, respectively.

*rrnB*P1:*tggcagttttaggctgatttggttgaatgttgcgcggtcagaaaattattt taaatttcctcttgtcaggccggaataactccctataatgcgcc***A**CCACTGACACG-GAACAACGGCAAACACGCCGCCGGGTCAGCGGGGTTCTCCTGAGA-A**C**TCCGGCAGAGAAAGCAAAAATAAATGC

*rrnB*P2:*cacgccgccgggtcagcggggttctcctgagaactccggcagagaaagcaa aaataaatgcttgactctgtagcgggaaggcgtattatgcacac***C**CCCGCGCCGCTGA-GAAAAAGCGAAGCGGCACTGCTCTTTAACAATTTATCA̲GACAATCTGT-GTGGGCACTCGAAGAT̲

A1:*tccagatcccgaaaatttatcaaaaagagtattgacttaaagtctaacctataggat acttacagcc*ATCGAGAGGGCCACGGCGAACAGCCAACCCAATCGAACAG-GCCTGCTGGTAATCGCA̲GGCCTTTTTATTT GGATCCCCGGGTA̲

To measure effect of CedA on ribosomal promoter initiation, 10 pmol of RNAP were mixed with either CedA up to 5 µM (WT or CedAΔ11N mutant as indicated) or equal amounts of TB100A in 60 µl of transcription buffer TB100A (40 mM Tris-HCl, pH 8.0, 100 mM NaCl, 10 mM MgCl$_2$; 0.1 mg ml$^{-1}$ BSA). Samples were incubated 5 min at 37 °C and split into two 30-µl parts each. One part from each set was mixed with DksA up to 0.5 µM and ppGpp up to 50 µM, and the second part was mixed with equal amount of TB100A. Samples were incubated at 37 °C for 5 min. Transcription was initiated with addition of 12 pmol *rrnB*P1 PCR promoter fragment premixed with 1 mM ATP and 4 µCi α-[$^{32}$P]-CTP (3,000 Ci mmol$^{-1}$; Perkin Elmer) at 37 °C. Next, 10-µl aliquots were withdrawn at 3, 6 or 12 min intervals and quenched in fresh tubes with 10 µl Stop Buffer (1× TBE buffer, 8 M urea, 20 mM EDTA, 0.025% xylene cyanole, 0.025% bromophenol blue). For the *rrnB*P2 promoter, the experiment was performed in the same way except *rrnB*P2 PCR promoter fragment was added together with 100 µM ApC RNA primer, 1 mM GTP and 4 µCi α-[$^{32}$P]-CTP. The products were separated on 23% polyacrylamide gel containing 8 M urea in TBE (20 × 20 cm) for 30 min at 50 W. The gel was transferred onto a film, covered with Saran wrap and exposed to a storage phosphor screen. The screen was scanned on Typhoon Imager (GE) and analyzed using Image Quant software (GE). Relative intensity values were retrieved directly from Image Quant software. The mean and standard deviation were calculated in Excel (Microsoft) after normalization and produced a less than 10% variation based on at least three independent experiments.

To measure open-promoter complex stability, 10 pmol of RNAP were mixed with equal amount of corresponding promoter DNA in 100 µl of TB100A and then split into two 50-µl aliquots each. The aliquots were mixed with either CedA up to 5 µM or equal amounts of TB100A, and incubated for 10 min at 37 °C. One 10 µl aliquot from each reaction was taken and chased in a new tube with either 1 mM ATP and 1 µCi α-[$^{32}$P]-CTP for *rrnB*P1 or 100 µM ApC RNA primer, 1 mM GTP and 1 µCi α-[$^{32}$P]-CTP for *rrnB*P2 promoter DNA for 5 min at 37 °C before quenching with 10 µl of Stop Buffer. The rest of the samples were mixed with heparin (up to 10 µg ml$^{-1}$) and 10 µl aliquots were withdrawn at 1, 2, 4 or 8 min intervals and placed into fresh tube with chase mixtures as described above. Samples were incubated 5 min at 37 °C and quenched with 10 µl of Stop Buffer. Samples were analyzed and visualized as described above.

To measure inhibition with rifampicin in the presence of CedA, 10 pmol of RNAP were mixed with equal amount of A1 PCR fragment in 200 µl of TB100A. The sample was split into two 100-µl parts: one aliquot was mixed with 5 pmol of CedA and the second with an equal amount of the same buffer. Samples were incubated for 5 min at 37 °C, and each was split into five 20 µl aliquots. Samples were mixed with

rifampicin solution (prepared fresh from dry powder; Sigma) up to 0.2; 0.5, 1 or 2 µg ml⁻¹ or equal amounts of water and incubated for 5 min at 37 °C. RNA synthesis was initiated by addition of 1 mM ATP, GTP, UTP and 10 µM CTP mixed with 0.5 µCi α-[$^{32}$P]-CTP. After incubation at 37 °C, 10-µl aliquots were withdrawn at 3- or 10-min intervals and quenched in new tubes with 10 µl of Stop Buffer. Samples were separated and visualized as described above except 30% polyacrylamide gel was used instead of 23%.

## qPCR

To measure cedA mRNA production, overnight culture of *E. coli* MG1655 was diluted 200 times with 80 ml of fresh LB media and the cells were grown in a 500 ml flask at 37 °C with shaking until the OD$_{600}$ 0.35. Cell culture was split into eight 10-ml parts in 40 ml glass tubes and samples were mixed with corresponding stressors: 2 mM H$_2$O$_2$, 50 µg ml⁻¹ kanamycin, 20 µg ml⁻¹ gentamycin, 50 µg ml⁻¹ erythromycin, 1 µg ml⁻¹ ciprofloxacin and 100 µg ml⁻¹ ampicillin. The shaking continued at 37 °C for 20 min. Cells were collected by 5 min of centrifugation at 5,000$g$ at 4 °C. For UV stress measurement, 10 ml of the cell culture was irradiated in a Petri dish with 64 J cm⁻² UV light (254 nm) at room temperature for 2 min. Cells were allowed to recover in darkness for 10 min at 37 °C before collecting.

mRNA was isolated using Master Pure Complete DNA&RNA purification kit (Biosearch technologies) according to the manufacturer's instructions except that scale was increased twice. Resulting RNA was dissolved in TE and concentration was adjusted to 500 ng ml⁻¹.

Complementary DNA (cDNA) was produced from 1 µg of total RNA (final cDNA concentration 5 ng ml⁻¹) using a QuantiTech Reverse Transcription kit (Qiagen) according to the manufacturer's instructions.

Quantitative PCR (qPCR) was performed with 2.5 ng of DNA in triplicates using PowerSYBR Green PCR master mix at Quant Studio v.7 qPCR machine (both Applied Biosystems) according to the manufacturer. Primer pairs used for qPCR for *cedA*, *katE* and *gapA* as a reference gene are shown below.

*cedA* or *katE* mRNA levels were normalized to *gapA* mRNA level produced from the same sample and resulting changes in Ct values (ΔCt) were compared between treated and untreated samples. −log$_2$ differences in Ct values (ΔΔCt) were plotted at $y$ axis as fold change.

qPCR with reverse transcription (RT–qPCR) primer GapA Forward GCACCACCAACTGCCTGGCT

RT–qPCR primer GapA Reverse CGCCGCGCCAGTCTTTGTGA
RT–qPCR primer CedA Forward CCGCCAGAACATGCGATAA
RT–qPCR primer CedA Reverse GCAGAAATCACTCTCCCATCAG
RT–qPCR primer KatE Forward CAGTCACCACTACACGATTCC
RT–qPCR primer KatE Revese CTGATTAGTGGTCAGCGCATAA

## Bacterial stress survival assay

Bacterial culture of *E. coli* MG1655 bearing paTc-*cedA* plasmid was grown overnight in LB media supplemented with 100 µg ml⁻¹ carbenicillin. The next day, 10 ml of LB media were inoculated with 20 µl (500× dilution) of the overnight culture and cells were grown at 37 °C with shaking for 1 h in a 40 ml glass tube. Sample was split into two 5 ml parts in 40 ml glass tubes and 100 ng ml⁻¹ anhydrotetracycline was added to one of them. Cultures were grown for 1 h at 37 °C with shaking. Samples were again diluted 1,000× in 10 ml of fresh LB media containing various stressors: 2 mM H$_2$O$_2$, 50 µg ml⁻¹ kanamycin, 20 µg ml⁻¹ gentamycin, 50 µg ml⁻¹ erythromycin and 1 µg ml⁻¹ ciprofloxacin. Control cultures with and without induction contained no stressors. Cells were allowed to grow for 90 min at 37 °C with shaking, spun down at 5,000$g$ for 5 min, media was discarded and cells were resuspended in 1 ml of sterile PBS at pH 7.2 (Gibco). Cells were serially diluted with PBS and plated at LB agar plates containing no antibiotics. Plates were grown overnight at 37 °C and colonies were counted.

## Reporting summary

Further information on research design is available in the Nature Portfolio Reporting Summary linked to this article.

## Data availability

NGS data are available at the Sequence Read Archive (https://www.ncbi.nlm.nih.gov/sra) with BioProject identifier PRJNA924329. MS data are available at PRIDE (https://www.ebi.ac.uk/pride/) with the project identifier PXD039446. Structure coordinates are available at the PDB (https://www.rcsb.org) with accession code 8FTD and at the Electron Microscopy Data Bank (EMDB) (https://www.ebi.ac.uk/emdb) with the code EMD-29423. Source data are provided with this paper.

## Code availability

Data analysis R scripts are available at https://doi.org/10.5281/zenodo.8357500.

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

## Acknowledgements

We thank W. Rice, B. Wang and A. Paquette for helping with sample screening and data collection at the NYULH (NYU Langone Health) Cryo-EM Laboratory. We thank the BigPurple HPC core at NYULH for computer access. This work was supported by the NIH grant no. R01 GM126891, Blavatnik Family Foundation, and by the Howard Hughes Medical Institute (E.N.).

## Author contributions

N.V. carried out all genetic, proteomics, and transcriptomics studies. M.M.J.L. purified the proteins, collected and processed EM data and built the structural model. V.E. performed in vitro transcription assays. I.S. performed the NGS. E.N. supervised the project. N.V. and E.N. wrote the manuscript with input from all the authors.

## Competing interests

The authors declare no competing interests.

## Additional information

**Extended data** is available for this paper at https://doi.org/10.1038/s41594-023-01154-w.

**Correspondence and requests for materials** should be addressed to Evgeny Nudler.

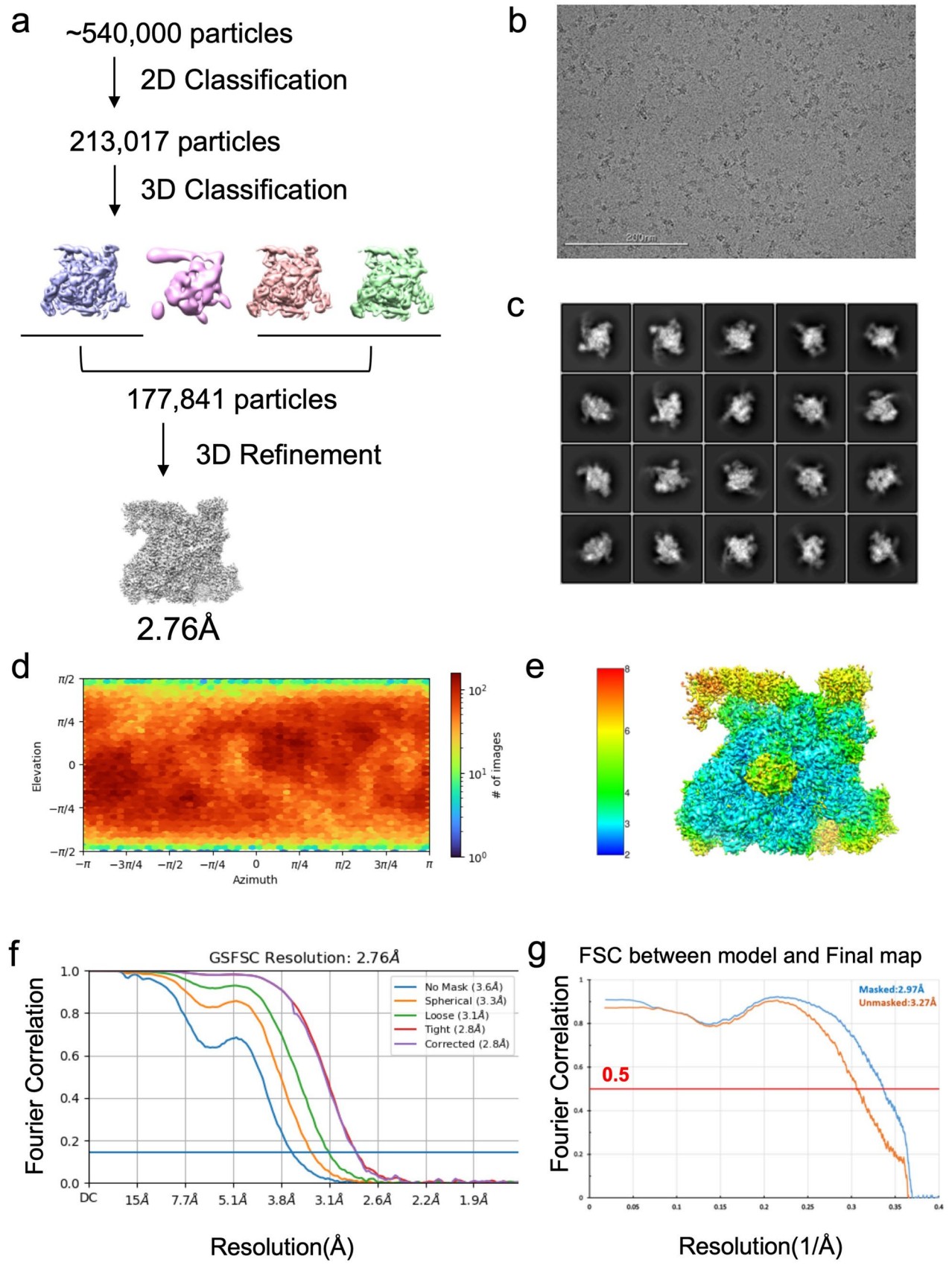

**Extended Data Fig. 1 | Cryo-EM determination of CedA bound to open-promoter complex. a**, Flowchart of cryo-EM data processing. **b**, Representative cryo-EM micrographs. Scale bar, 200 nm. **c**, Representative 2D class average shows different views, suggesting randomly distributed particles. **d**, Angular distribution; **e**, local ResMap estimation; **f**, Fourier Shell Correlation (FSC) curves of final reconstruction; **g**, FSC plots of final reconstruction against the model.

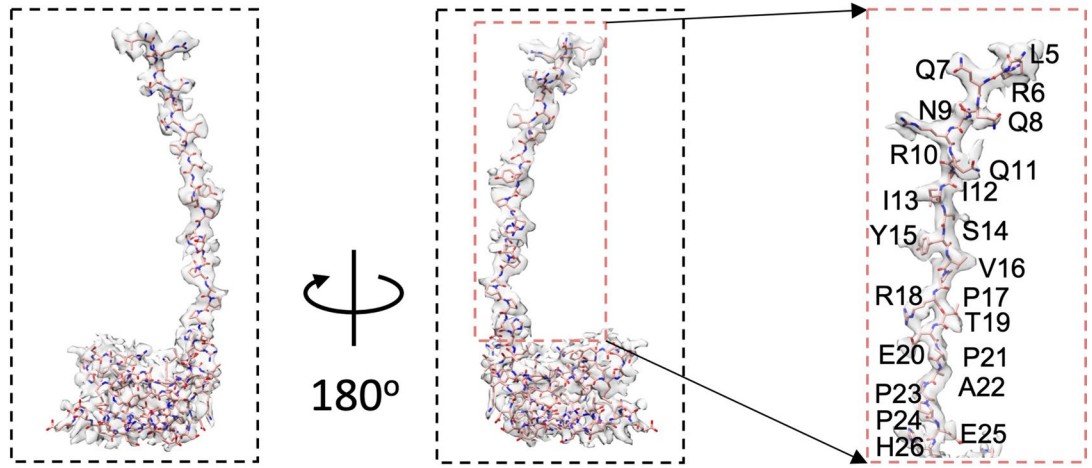

**Extended Data Fig. 2 | Cryo-EM density map of CedA.** The electron density extracted from the 2.76 Å cryo-EM map shows the position of CedA side chains.

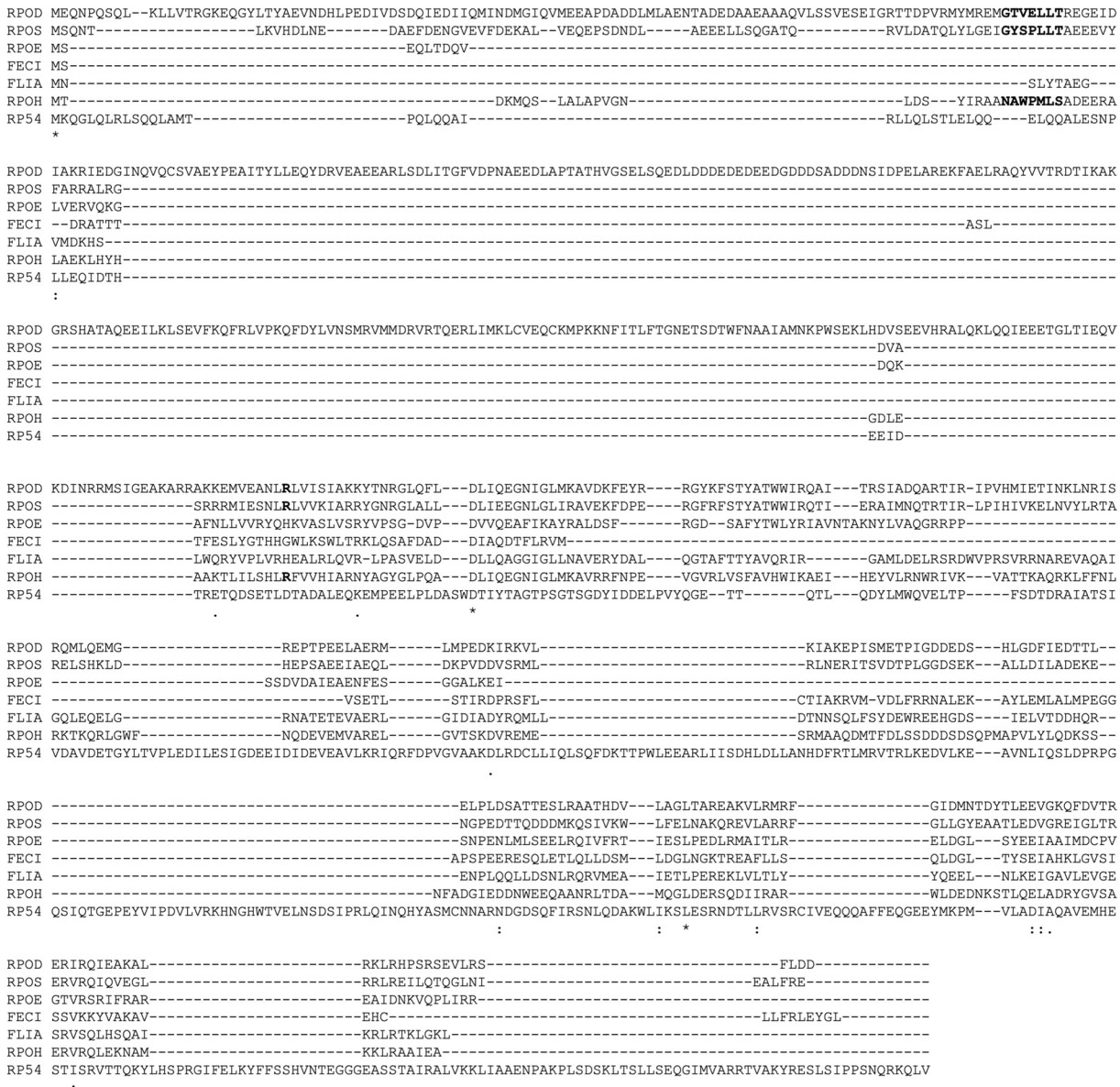

**Extended Data Fig. 3 | Sequence alignment of *E. coli* sigma factors.** Amino acid residues participating in interaction with CedA are shown in bold.

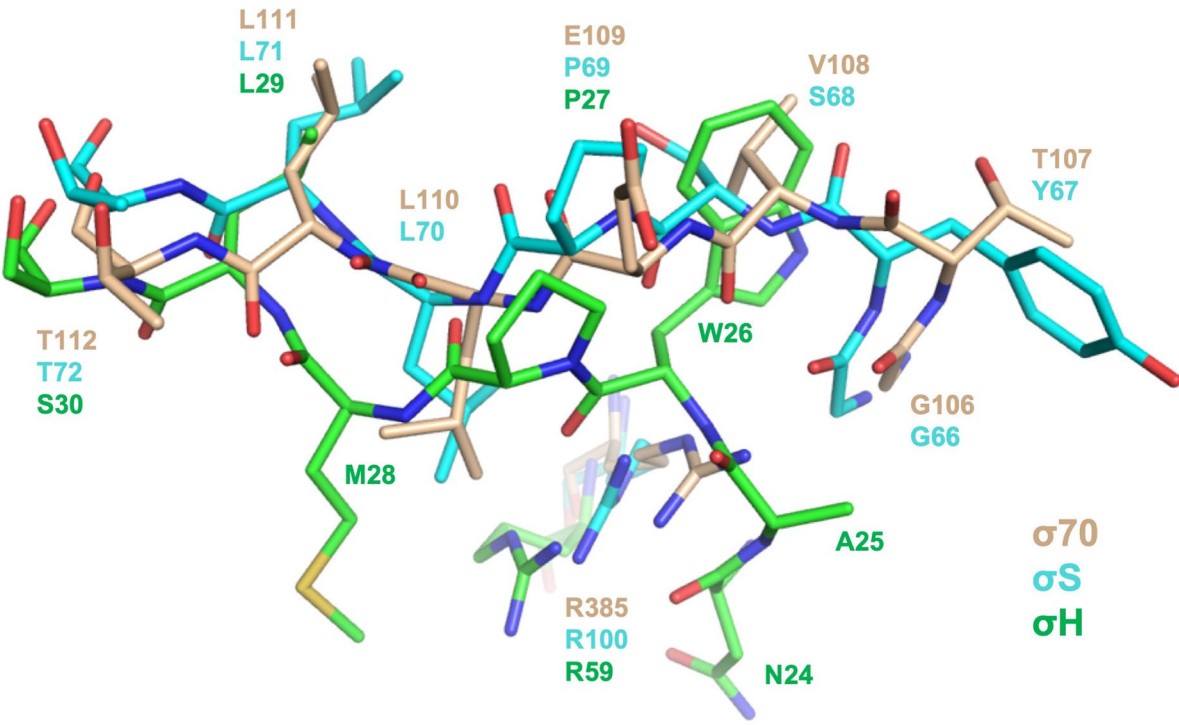

**Extended Data Fig. 4 | Structural comparison of region 1.2 in σ70, σS and σH.** Structure models of σ[70] (shown in wheat, this study), σ[S] (shown in cyan, PDB: 6OMF), and σ[H] (shown in green, AlphaFold: AF-P0AGB3-F1) were aligned using PyMol 2.4. Region 1.2 interacting with CedA is shown.

*Double-anonymous peer review submissions: write DAPR and your manuscript number here instead of author names.*

## Reporting Summary

### Statistics

For all statistical analyses, confirm that the following items are present in the figure legend, table legend, main text, or Methods section.

| n/a | Confirmed | |
|---|---|---|
| ☐ | ☒ | The exact sample size (*n*) for each experimental group/condition, given as a discrete number and unit of measurement |
| ☐ | ☒ | A statement on whether measurements were taken from distinct samples or whether the same sample was measured repeatedly |
| ☐ | ☒ | The statistical test(s) used AND whether they are one- or two-sided *Only common tests should be described solely by name; describe more complex techniques in the Methods section.* |
| ☐ | ☒ | A description of all covariates tested |
| ☐ | ☒ | A description of any assumptions or corrections, such as tests of normality and adjustment for multiple comparisons |
| ☐ | ☒ | A full description of the statistical parameters including central tendency (e.g. means) or other basic estimates (e.g. regression coefficient) AND variation (e.g. standard deviation) or associated estimates of uncertainty (e.g. confidence intervals) |
| ☐ | ☒ | For null hypothesis testing, the test statistic (e.g. *F*, *t*, *r*) with confidence intervals, effect sizes, degrees of freedom and *P* value noted *Give P values as exact values whenever suitable.* |
| ☒ | ☐ | For Bayesian analysis, information on the choice of priors and Markov chain Monte Carlo settings |
| ☒ | ☐ | For hierarchical and complex designs, identification of the appropriate level for tests and full reporting of outcomes |
| ☒ | ☐ | Estimates of effect sizes (e.g. Cohen's *d*, Pearson's *r*), indicating how they were calculated |

*Our web collection on statistics for biologists contains articles on many of the points above.*

### Software and code

Policy information about availability of computer code

| Data collection | Mass spectrometry data were collected using Thermo Scientific Xcalibur v4.1. |
|---|---|
| Data analysis | Mass spectrometry data were analyzed using MaxQuant v2.0.1. Next-generation sequencing data were analyzed using Bowtie2 v2.4, Samtools v1.13 and custom R scripts. Statistical analysis was performed using R v4.1, including packages Rsubread, vsn, qvalue. Sequence alignment was done using MAFFT v7.487. Leginon v1.0 was used to collect electron microphotographs. Cryo-EM data were processed using CTFFIND4 v4.0.8, Gautomatch v0.56, RELION v3.1.2, cryoSPARC v3.2.0. Structure visualization was done in Chimera v1.16, map resolution was calculated by ResMap v1.1.4. Model was build in COOT v0.9.8 and refined in Phenix v1.18.2. |

For manuscripts utilizing custom algorithms or software that are central to the research but not yet described in published literature, software must be made available to editors and reviewers. We strongly encourage code deposition in a community repository (e.g. GitHub). See the Nature Portfolio guidelines for submitting code & software for further information.

## Data

Policy information about <u>availability of data</u>

All manuscripts must include a <u>data availability statement</u>. This statement should provide the following information, where applicable:

- Accession codes, unique identifiers, or web links for publicly available datasets
- A description of any restrictions on data availability
- For clinical datasets or third party data, please ensure that the statement adheres to our <u>policy</u>

Mass spectrometry data is available at PRIDE database with project ID PXD039446.
Next-generation sequencing data is available at SRA with BioProject accession number PRJNA924329.
Structure coordinates are available at PDB with accession code 8FTD and at EMDB with code EMD-29423.

## Research involving human participants, their data, or biological material

Policy information about studies with <u>human participants or human data</u>. See also policy information about <u>sex, gender (identity/presentation), and sexual orientation</u> and <u>race, ethnicity and racism</u>.

| | |
|---|---|
| Reporting on sex and gender | Not applicable |
| Reporting on race, ethnicity, or other socially relevant groupings | Not applicable |
| Population characteristics | Not applicable |
| Recruitment | Not applicable |
| Ethics oversight | Not applicable |

Note that full information on the approval of the study protocol must also be provided in the manuscript.

# Field-specific reporting

Please select the one below that is the best fit for your research. If you are not sure, read the appropriate sections before making your selection.

☒ Life sciences    ☐ Behavioural & social sciences    ☐ Ecological, evolutionary & environmental sciences

For a reference copy of the document with all sections, see <u>nature.com/documents/nr-reporting-summary-flat.pdf</u>

# Life sciences study design

All studies must disclose on these points even when the disclosure is negative.

| | |
|---|---|
| Sample size | Study was done with bacterial cultures that were used in their entirety for high-throughput experiments. Volume of cultures was chosen to provide enough material (protein, protein-bound DNA, or RNA) for analyses, based on published estimates of corresponding molecules copy number (for reference see https://doi.org/10.1038/nbt.3418, https://www.thermofisher.com/us/en/home/references/ambion-tech-support/rna-tools-and-calculators/macromolecular-components-of-e.html). For plating experiments, bacterial cultures were diluted so that individual colonies could be grown and counted. |
| Data exclusions | No data were excluded from analyses. |
| Replication | Experiments were performed in replicates on different days. Two replicates were done for ChIP-seq experiment, three replicates for RNA-seq and LC-MS experiments. |
| Randomization | Not applicable. This is not a confirmatory nor clinical study, and does not involve animals or humans. |
| Blinding | Not applicable. This is not a confirmatory nor clinical study, and does not involve animals or humans. |

# Reporting for specific materials, systems and methods

We require information from authors about some types of materials, experimental systems and methods used in many studies. Here, indicate whether each material, system or method listed is relevant to your study. If you are not sure if a list item applies to your research, read the appropriate section before selecting a response.

## Materials & experimental systems

| n/a | Involved in the study |
|---|---|
| ☐ | ☒ Antibodies |
| ☒ | ☐ Eukaryotic cell lines |
| ☒ | ☐ Palaeontology and archaeology |
| ☒ | ☐ Animals and other organisms |
| ☒ | ☐ Clinical data |
| ☒ | ☐ Dual use research of concern |
| ☒ | ☐ Plants |

## Methods

| n/a | Involved in the study |
|---|---|
| ☐ | ☒ ChIP-seq |
| ☒ | ☐ Flow cytometry |
| ☒ | ☐ MRI-based neuroimaging |

## Antibodies

| | |
|---|---|
| Antibodies used | Pierce anti-DYKDDDDK antibody covalently coupled to Magnetic Agarose (cat # A36797). |
| Validation | Per manufacturers (Thermo Scientific) manual, the product is high-affinity rat monoclonal antibody (clone L5) that is covalently attached to a magnetite-embedded agarose core particle with binding capacity of equal or more than 3.2 mg of DYKDDDDK-tGFP-His protein (32 kDa) per 1 ml of settled beads. Manufacturer demonstrated use of affinity resin for isolation of various N- and C-terminally FLAG-tagged proteins with high yield and purity. |

## Plants

| | |
|---|---|
| Seed stocks | Not applicable |
| Novel plant genotypes | Not applicable |
| Authentication | Not applicable |

## ChIP-seq

### Data deposition

☒ Confirm that both raw and final processed data have been deposited in a public database such as GEO.

☒ Confirm that you have deposited or provided access to graph files (e.g. BED files) for the called peaks.

| | |
|---|---|
| Data access links *May remain private before publication.* | https://www.ncbi.nlm.nih.gov/sra/?term=PRJNA924329 |
| Files in database submission | ctl_1_R1.fastq.gz, ctl_1_R2.fastq.gz, ctl_2_R1.fastq.gz, ctl_2_R2.fastq.gz, ceda_1_R1.fastq.gz, ceda_1_R2.fastq.gz, rpoc_1_R1.fastq.gz, rpoc_1_R2.fastq.gz, rpoc_2_R1.fastq.gz, rpoc_2_R2.fastq.gz |
| Genome browser session (e.g. UCSC) | Not applicable. |

### Methodology

| | |
|---|---|
| Replicates | Two technical replicates for each bacterial strain were done on different days. |
| Sequencing depth | 75-nt paired reads. experiment, total reads, uniquely mapped (-F 4 -q 30) ctl_1, 32543296, 31289041 ceda_1, 25728884, 24639037 rpoc_1, 37064686, 21616956 ctl_2, 29238808, 27442010 ceda_2, 26947978, 25521810 rpoc_2, 38674750, 23064615 |
| Antibodies | Pierce™ Anti-DYKDDDDK Magnetic Agarose, Thermo Scientific Cat A36797. |
| Peak calling parameters | 3 sigma above background level measured in control samples (IP from FLAG-less E. coli strain). |
| Data quality | Only properly mapped pairs of reads with mapping quality above 10 were considered for analysis. |
| Software | For reads mapping and filtering bowtie2 and samtools were used. Peak calling was done in R. R scripts are provided with submission. |

