## [Peer Review File · Nature Structural & Molecular Biology]

Peer Review Information

Manuscript Title: General Transcription Factor from E. coli with a Distinct Mechanism of Action

Corresponding author name(s): Evgeny Nudler

Reviewer Comments & Decisions:

Decision Letter, initial version:

Message: 21st Jul 2023

Dear Professor Nudler,

Thank you again for submitting your manuscript "General Transcription Factor from E. coli with a Distinct Mechanism of Action". I apologise for the delay in responding, which resulted from the difficulty in timely obtaining suitable referee reports. Nevertheless, we now have comments (below) from the 2 reviewers who evaluated your paper. In light of those reports, we remain interested in your study and would like to see your response to the comments of the referees, in the form of a revised manuscript.

You will see that both experts appreciate both the orthogonality of the approaches employed and the mechanistic/functional insight imparted by this work. However, the experts voice some concerns with respect to parts of the manuscript that are unclear or confusing. We ask that you take heed of these requests and provide all necessary clarifications. Moreover, the reviewers pose certain mechanistic questions (Reviewer#1: major point and Reviewer#2: points 8 and 9). We editorially agree that addressing these questions will further strengthen this manuscript. As always we ask you to fully address all the points raised by the referees in a point-by-point response.

We expect to see your revised manuscript within 3 months. If you cannot send it within this time, please contact us to discuss an extension; we would still consider your revision, provided that no similar work has been accepted for publication at NSMB or published elsewhere.

As you already know, we put great emphasis on ensuring that the methods and statistics

reported in our papers are correct and accurate. As such, if there are any changes that should be reported, please submit an updated version of the Reporting Summary along with your revision.

Reporting Summary:

Data availability: this journal strongly supports public availability of data. All data used in accepted papers should be available via a public data repository, or alternatively, as Supplementary Information. If data can only be shared on request, please explain why in your Data Availability Statement, and also in the correspondence with your editor. Please note that for some data types, deposition in a public repository is mandatory - more information on our data deposition policies and available repositories can be found below: <https://www.nature.com/nature-research/editorial-policies/reporting-standards#availability-of-data>

[redacted]

Sincerely,

Dimitris Typas
Associate Editor
Nature Structural & Molecular Biology
ORCID: 0000-0002-8737-1319

Referee expertise:

Referee #1: structure-function characterization of bacterial transcription complexes

Referee #2: structural characterization of bacterial transcription complexes

Reviewers' Comments:

Reviewer #1:

Remarks to the Author:

This manuscript describes a comprehensive analysis of CedA, an *E. coli* protein described as a regulator of cell division ~25 years ago. CedA has been studied by several groups and its structure revealed a DNA-binding domain; furthermore, CedA interactions with DNA and RNA polymerase have been reported, suggesting that it acts as a transcription factor, but the mechanism of CedA remained elusive. In this work, Evgeny Nudler's lab used proteomics, genome-wide analysis of CedA localization on the chromosome and its effects on gene expression, cryo-EM analysis of CedA-bound open promoter complex, and in vitro biochemistry to show that CedA is a novel type of global transcription factor that binds to RNA polymerase, rather than to DNA, and alters the transcription complex properties. This mechanism is conceptually analogous to that of DksA, a cofactor of the stress alarmone ppGpp; DksA has a zinc-finger domain and has been initially thought to bind to promoter DNA and act as a canonical transcription initiation factor. The authors' structure, obtained on a promoter they identified as a target of CedA, demonstrates that CedA bridges two distant domains of RNA polymerase beta and sigma subunits, locking the enzyme in a state resistant to conformational changes induced by DksA (and similar factors). This observation makes predictions about the interplay between CedA and DksA and the effects of CedA on open complex stability, which the authors confirmed by in vitro biochemistry.

The authors also show that the expression of *cedA*, which is divergently transcribed from a promoter that overlaps that of *katE*, which encodes a catalase, is activated by stresses including hydrogen peroxide, UV, and aminoglycoside antibiotics; in turn, overexpression of *cedA* confers tolerance to these stresses, as well as to rifampicin.

Studies of proteins that control transcription through direct contacts to RNA polymerase are complicated because they have many hundreds of targets and exert subtle, but physiologically significant, effects on expression of numerous genes. While the authors cannot identify the "key" targets of CedA, it is clearly a global regulator with important effects on bacterial pathogenesis and fitness. The notion that non-DNA binding protein regulators like CedA (DksA, Suta, and others) and small molecules (like ppGpp) have pleiotropic effects is very important to stress – there are many similar factors that remain understudied owing to the lack of simple assays.

In summary, this is a compelling multi-pronged analysis of a new transcription regulator. The conclusions are supported by the data, the manuscript is clearly written and concise, and the figures are easy follow.

That said, I found the last section of the results, describing Fig. 6c-e, to be confusing. Fig. 6c shows that, in the absence of rifampicin, a very large number of genes is affected by the deletion of *cedA*, and the effects are generally small and reversed upon CedA overexpression, as could be expected. In the presence of rifampicin, the wt and deletion strains are also quite similar yet, upon rifampicin addition, the picture dramatically changes – not only more genes are affected, but the pattern does not appear to revert to the wt strain on steroids, consistent with lots of indirect effects, as the authors pointed out. However, the look at "cognate" promoters where CedA could have direct effects (Fig. 6e) on gene expression is not very helpful – since there are only 10, why not list them in a table?

Otherwise, I have only a few minor suggestions:

Add a clear statement that CedA was among the top 20 hits to the end of the proteomics paragraph, on line 109. I would add that part of Table S2 to figure 1. Perhaps add a

statement about CedA abundance? Relatively low numbers have been estimated from ribosome profiling, but these are indirect measurements.

Revise the paragraph that starts at line 175. A better description of DksA/TraR conformational changes may be useful, perhaps with a schematic. And RNA polymerase does not have a major groove (or a minor one).

OPC is not a common name for the open promoter complex, and rrnBP1/P2 are used instead of p1 and P2. Why not use the standard names?

Line 298 delete either

Line 301 delete TraR

Fig. 2a: RNAP trace is cyan, not green, which is better anyway

Fig. 4 can be improved. Remove outside lines in panels b and d, they look messy. Also, although the results are clear and quantification is shown, some indication of data variability and the number of experiments should be given in the figure legend, not in the Methods section where it is now. Based on the timecourses in Fig. 4c, the errors must be more than 10% - a better description how the data were normalized is needed. It is also unclear what the numbers refer to.

Fig. 5: does CedA bind to its own promoter? In other words, are its effects in part mediated by increased katE expression? In panel c, show only even numbers on the y-axis.

Fig. 6b: same comment as for Fig. 4c.

Reviewer #2:

Remarks to the Author:

Vassyliev et al. present a combined multi-omics, structural and biochemical study to elucidate the cellular functions and molecular mechanism of the CedA protein in *E. coli*, which had previously been suggested to act as an activator of cell division.

The authors set out to delineate the cellular protein interactome of CedA by conducting crosslinking/MS in cells, identifying sigma70 holoenzyme as a main interactor. Using ChIP-seq, they revealed proximity of CedA to -10 promoter elements. They elucidated a high-resolution cryoEM structure of a CedA-modified sigma70 open promoter complex (OPC) assembled on the promoter for which the strongest CedA association had been observed. Surprisingly, CedA did not directly contact the DNA as previously expected but used a C-terminal domain resembling double stranded DNA binding proteins to engage lineage-specific insertion beta_i4. An extended, intrinsically disordered N-terminal region of CedA bridges the beta lobe and sigma 70 region 1.2. Structural comparisons suggested that CedA might prevent a conformational change in the beta lobe and beta_i4 observed in DksA- or TraR-modified initiation complexes. Consistently, in vitro transcription assays revealed that CedA (but not a variant lacking the sigma70-contacting N-terminal region) counteracts DksA effects and stabilizes OPCs. Based on its genomic location, the authors hypothesized possible cellular functions of CedA and showed that cedA expression is up-regulated under oxidative stress and in the presence of certain antibiotics, and that CedA increases cell survival in the presence of these stressors. They also observed increased tolerance of cells against rifampicin in the presence of CedA but not in presence of CedA lacking the sigma70-contacting N-terminal region. Finally, comparative RNA-seq with wt *E. coli* or strains lacking or overproducing CedA revealed many genes differentially modulated by CedA and rifampicin. Interestingly, while CedA predominantly represses cognate promoters under normal growth conditions, it predominantly activates a larger number of such promoters in the presence of rifampicin.

This is a carefully conducted, comprehensive study. The experimental work appears to be of high technical quality. The results are novel and interesting. They define CedA as a newly recognized transcription initiation factor that regulates a large number of genes in *E. coli*. Furthermore, the results pinpoint pivotal cellular roles of CedA in mitigating effects of oxidative and antibiotics stress. Finally, the study convincingly links CedA function to its molecular mechanism as a modulator of transcription initiation that stabilizes OPCs.

Specific comments:

1. Line 162: "major groove" should rather be "main channel".
2. Figure 4a: the structural comparisons are not entirely clear. It seems as if the authors present "hybrid" structures with SspA or DksA/beta lobe transplanted onto the CedA complex? For the very least, this should be clearly described in the legend. Perhaps only the relevant elements/components (CedA, SspA, DksA, lobe) should be highlighted in color with the remainder in gray?
3. Concerning SspA – according to Figure 1c, SspA is also a CedA interactor? Do the authors concern any conflict in SspA and CedA binding concomitantly? Would they expect opposing effects as for CedA/DksA or perhaps even synergistic effects? They should briefly refer to the possible SspA-CedA interplay in the manuscript.
2. Figure 4b-d and Figure 6b: While effects are clear, how were values referred to as "intensity" evaluated; what do the numbers represent?
3. Figure 4b: CedA alone activates *rrnBp1* but represses *rrnBp2*. Do the authors have an explanation for the opposing effects of CedA on different promoters?
4. Figure 5b-d: the authors should indicate the number and type (biological, technical) of replicates.
5. Figure 5b,c: indicate on y-axes which mRNA fold change is shown (*katA*, *cedA*)
6. Figure 5c: does rifampicin also enhance expression of *cedA*?
7. The authors should show an exemplary region of the cryoEM density in detail. One possibility would be Fig. 3c (density of the CedA region instead of the surface of the CedA model).
8. Figure 6e: upon CedA overproduction - which fraction of the promoters that are down-regulated under control conditions are up-regulated upon treatment with rifampicin? If a significant number of promoters switch from down- to up-regulation, how could this be explained based on the CedA mode of action during initiation? While a possible explanation (stabilization of OPC/delayed promoter clearance) is provided for CedA-mediated down-regulation, how about CedA-mediated up-regulation, and how could the effect switch under rifampicin treatment?
9. Considering its originally suggested function as an activator of cell division, can the author discern any possible explanation for this role of CedA? For instance, are there any

CedA-dependent promoters they identified that control genes whose products might affect cell division?

Author Rebuttal to Initial comments

Reviewer #1:

This manuscript describes a comprehensive analysis of CedA, an *E. coli* protein described as a regulator of cell division ~25 years ago. CedA has been studied by several groups and its structure revealed a DNA-binding domain; furthermore, CedA interactions with DNA and RNA polymerase have been reported, suggesting that it acts as a transcription factor, but the mechanism of CedA remained elusive. In this work, Evgeny Nudler's lab used proteomics, genome-wide analysis of CedA localization on the chromosome and its effects on gene expression, cryo-EM analysis of CedA-bound open promoter complex, and in vitro biochemistry to show that CedA is a novel type of global transcription factor that binds to RNA polymerase, rather than to DNA, and alters the transcription complex properties. This mechanism is conceptually analogous to that of DksA, a cofactor of the stress alarmone ppGpp; DksA has a zinc-finger domain and has been initially thought to bind to promoter DNA and act as a canonical transcription initiation factor. The authors' structure, obtained on a promoter they identified as a target of CedA, demonstrates that CedA bridges two distant domains of RNA polymerase beta and sigma subunits, locking the enzyme in a state resistant to conformational changes induced by DksA (and similar factors). This observation makes predictions about the interplay between CedA and DksA and the effects of CedA on open complex stability, which the authors confirmed by in vitro biochemistry.

The authors also show that the expression of *cedA*, which is divergently transcribed from a promoter that overlaps that of *katE*, which encodes a catalase, is activated by stresses including hydrogen peroxide, UV, and aminoglycoside antibiotics; in turn, overexpression of *cedA* confers tolerance to these stresses, as well as to rifampicin.

Studies of proteins that control transcription through direct contacts to RNA polymerase are complicated because they have many hundreds of targets and exert subtle, but physiologically significant, effects on expression of numerous genes. While the authors cannot identify the "key" targets of CedA, it is clearly a global regulator with important effects on bacterial pathogenesis and fitness. The notion that non-DNA binding protein regulators like CedA (DksA, Suta, and others) and small molecules (like ppGpp) have pleiotropic effects is very important to stress – there are many similar factors that remain understudied owing to the lack of simple assays.

In summary, this is a compelling multi-pronged analysis of a new transcription regulator. The conclusions are supported by the data, the manuscript is clearly written and concise, and the figures are easy follow.

We thank the reviewer for their helpful comments and their enthusiasm for the novel findings of CedA. We have resolved all the issues with figure presentation and clarity as detailed below.

That said, I found the last section of the results, describing Fig. 6c-e, to be confusing. Fig. 6c shows that, in the absence of rifampicin, a very large number of genes is affected by the deletion of *cedA*, and the effects are generally small and reversed upon CedA overexpression, as could be expected. In the presence of rifampicin, the wt and deletion strains are also quite similar yet, upon rifampicin addition, the picture dramatically changes – not only more genes are affected, but the pattern does not appear to revert to the wt strain on steroids, consistent with lots of indirect effects, as the authors pointed out. However, the look at "cognate" promoters where

CedA could have direct effects (Fig. 6e) on gene expression is not very helpful – since there are only 10, why not list them in a table?

We appreciate the reviewer’s feedback and we have modified Fig. 6 and the text for greater clarity. Fig. 6c now includes hierarchical clustering dendrogram highlighting larger overall effect of *cedA* overexpression compared to its deletion in either control or rifampicin conditions. Ten “cognate” promoters that may specifically respond to CedA are now listed in Fig. 6e together with the effect size upon treatment with rifampicin in deletion and overexpression strains.

Otherwise, I have only a few minor suggestions:

Add a clear statement that CedA was among the top 20 hits to the end of the proteomics paragraph, on line 109. I would add that part of Table S2 to figure 1. Perhaps add a statement about CedA abundance? Relatively low numbers have been estimated from ribosome profiling, but these are indirect measurements.

Statement on CedA abundance and its position in rank list of proteins co-isolated with RpoC added to the text.

Figure 1c already lists 20 top-most ranking proteins co-isolated with RpoC, thus, adding mentioned part of Table S2 seems redundant.

Revise the paragraph that starts at line 175. A better description of DksA/TraR conformational changes may be useful, perhaps with a schematic. And RNA polymerase does not have a major groove (or a minor one).

Fixed

OPC is not a common name for the open promoter complex, and rrnBP1/P2 are used instead of p1 and P2. Why not use the standard names?

We now use the standard names throughout the revised manuscript.

Line 298 delete either

Done

Line 301 delete TraR

Done

Fig. 2a: RNAP trace is cyan, not green, which is better anyway

Corrected

Fig. 4 can be improved. Remove outside lines in panels b and d, they look messy. Also, although the results are clear and quantification is shown, some indication of data variability and the number of experiments should be given in the figure legend, not in the Methods section where it

is now. Based on the time courses in Fig. 4c, the errors must be more than 10% - a better description how the data were normalized is needed. It is also unclear what the numbers refer to.

We have revised the figure and its legend accordingly.

Fig. 5: does CedA bind to its own promoter? In other words, are its effects in part mediated by increased katE expression? In panel c, show only even numbers on the y-axis.

In tested conditions, CedA does not appear to bind to its own promoter. The CedA ChIP-seq signal at cedA promoter does not differ much from background.

Fig. 6b: same comment as for Fig. 4c.

Done

Reviewer #2: Vassyliov et al. present a combined multi-omics, structural and biochemical study to elucidate the cellular functions and molecular mechanism of the CedA protein in *E. coli*, which had previously been suggested to act as an activator of cell division.

The authors set out to delineate the cellular protein interactome of CedA by conducting crosslinking/MS in cells, identifying sigma70 holoenzyme as a main interactor. Using ChIP-seq, they revealed proximity of CedA to -10 promoter elements. They elucidated a high-resolution cryoEM structure of a CedA-modified sigma70 open promoter complex (OPC) assembled on the promoter for which the strongest CedA association had been observed. Surprisingly, CedA did not directly contact the DNA as previously expected but used a C-terminal domain resembling double stranded DNA binding proteins to engage lineage-specific insertion betaI4. An extended, intrinsically disordered N-terminal region of CedA bridges the beta lobe and sigma 70 region 1.2. Structural comparisons suggested that CedA might prevent a conformational change in the beta lobe and betaI4 observed in DksA- or TraR-modified initiation complexes. Consistently, in vitro transcription assays revealed that CedA (but not a variant lacking the sigma70-contacting N-terminal region) counteracts DksA effects and stabilizes OPCs. Based on its genomic location, the authors hypothesized possible cellular functions of CedA and showed that cedA expression is up-regulated under oxidative stress and in the presence of certain antibiotics, and that CedA increases cell survival in the presence of these stressors. They also observed increased tolerance of cells against rifampicin in the presence of CedA but not in presence of CedA lacking the sigma70-contacting N-terminal region. Finally, comparative RNA-seq with wt *E. coli* or strains lacking or overproducing CedA revealed many genes differentially modulated by CedA and rifampicin. Interestingly, while CedA predominantly represses cognate promoters under normal growth conditions, it predominantly activates a larger number of such promoters in the presence of rifampicin.

This is a carefully conducted, comprehensive study. The experimental work appears to be of high technical quality. The results are novel and interesting. They define CedA as a newly recognized transcription initiation factor that regulates a large number of genes in *E. coli*. Furthermore, the results pinpoint pivotal cellular roles of CedA in mitigating effects of oxidative and antibiotics

stress. Finally, the study convincingly links CedA function to its molecular mechanism as a modulator of transcription initiation that stabilizes OPCs.

We thank this reviewer for their helpful comments and agreement that our work is novel and interesting and of high technical quality. We have addressed all of the reviewer's points as indicated below.

Specific comments:

1. Line 162: "major groove" should rather be "main channel". Corrected
2. Figure 4a: the structural comparisons are not entirely clear. It seems as if the authors present "hybrid" structures with SspA or DksA/beta lobe transplanted onto the CedA complex? For the very least, this should be clearly described in the legend. Perhaps only the relevant elements/components (CedA, SspA, DksA, lobe) should be highlighted in color with the remainder in gray?

Done

3. Concerning SspA – according to Figure 1c, SspA is also a CedA interactor? Do the authors concern any conflict in SspA and CedA binding concomitantly? Would they expect opposing effects as for CedA/DksA or perhaps even synergistic effects? They should briefly refer to the possible SspA-CedA interplay in the manuscript.

SspA does not interact with CedA directly, as shown on figure 4a, but binds to $\sigma 70$ RNAP holoenzyme (<https://doi.org/10.1093/nar/gkaa672>). Thus, both CedA and SspA can form complex with RNAP• $\sigma 70$ independently and, potentially, work synergistically, however, this is out of scope of this study.

Based on relatively high abundance of SspA (more than 2-fold that of $\sigma 70$) measured in LB-grown *E. coli* (<https://doi.org/10.1038/nbt.3418>), it may very well occupy all available RNAP• $\sigma 70$ in the cell, thus, appear in both RpoC and CedA pulldowns.

2. Figure 4b-d and Figure 6b: While effects are clear, how were values referred to as "intensity" evaluated; what do the numbers represent?

We now provide the description of the numbers and statistics in the corresponding legends.

3. Figure 4b: CedA alone activates *rrnBp1* but represses *rrnBp2*. Do the authors have an explanation for the opposing effects of CedA on different promoters?

We describe possible explanation of opposing effect of CedA on *rrnBp1* and *rrnBp2* in Results section "CedA stabilizes an open-promoter complex and counteracts DksA". Briefly, increased life time of open promoter complex in the presence of CedA slows down promoter clearance by RNAP leading to diminished overall amount of synthesized RNA in multi-round transcription assay.

4. Figure 5b-d: the authors should indicate the number and type (biological, technical) of replicates.

Done

5. Figure 5b,c: indicate on y-axes which mRNA fold change is shown (katA, ceda)

Done

6. Figure 5c: does rifampicin also enhance expression of ceda?

RNA-seq data show no significant change in *cedA* mRNA abundance. LC-MS did not show change in Ceda abundance upon treatment with rifampicin either (LC-MS results are not included in the manuscript).

7. The authors should show an exemplary region of the cryoEM density in detail. One possibility would be Fig. 3c (density of the Ceda region instead of the surface of the Ceda model).

Following the reviewer's advice, we have enclosed a new supplementary figure (Fig. S4) to show the local density details of Ceda.

8. Figure 6e: upon Ceda overproduction - which fraction of the promoters that are down-regulated under control conditions are up-regulated upon treatment with rifampicin? If a significant number of promoters switch from down- to up-regulation, how could this be explained based on the Ceda mode of action during initiation? While a possible explanation (stabilization of OPC/delayed promoter clearance) is provided for Ceda-mediated down-regulation, how about Ceda-mediated up-regulation, and how could the effect switch under rifampicin treatment?

Fig. 6e has been modified as suggested by reviewer #1 to list 10 promoters that were downregulated in rifampicin-treated $\Delta cedA$ cells, of which 8 were upregulated upon *cedA* overexpression in the presence of rifampicin.

Stabilization of open promoter complex by Ceda may have opposite effects depending on promoter, particularly sequence and length of its discriminator region, as in case of *rrnB* P1 and P2, propensity to abortive initiation, etc.

Regarding the activity of promoters in the *cedA* overexpression strain, of 64 differentially regulated promoters only 13 were found to switch from downregulated (q-value < 0.01 and log₂ fold change below -0.5) in non-treated cells to upregulated (q-value < 0.01 and log₂ fold change above 0.5) in rifampicin-treated cells. The apparent switch from down- to upregulation may result from indirect effects such as redistribution of RNAP on chromosome (<https://doi.org/10.1046/j.1365-2958.2003.03805.x>), change of DNA topology (<https://doi.org/10.3389/fmicb.2015.00636>) that are known to occur upon treatment with rifampicin, or activity of other transcription factors responding to rifampicin rather than direct activity of Ceda.

9. Considering its originally suggested function as an activator of cell division, can the author discern any possible explanation for this role of CedA? For instance, are there any CedA-dependent promoters they identified that control genes whose products might affect cell division?

Our RNA-seq experiment identified hundreds of genes that were affected by *cedA* deletion and/or overexpression. Among putative candidates are indeed the genes encoding proteins known to participate or modulate cell division: *damX*, *dicB*, *FtsE*, *YmfM*. Besides, activation of cell division by CedA was described for *E. coli* bearing *dnaAcos* mutation, which by itself may skew gene expression pattern in a way unknown to us when grown in non-permissive conditions.

Decision Letter, first revision:

Message: Our ref: NSMB-A47694A

23rd Aug 2023

Dear Dr. Nudler,

Thank you for submitting your revised manuscript "General Transcription Factor from *E. coli* with a Distinct Mechanism of Action" (NSMB-A47694A). It has now been seen by the original referees and their comments are below. The reviewers find that the paper has improved in revision, and therefore we'll be happy to accept it in principle in Nature Structural & Molecular Biology, pending minor revisions to satisfy the referees' final requests and to comply with our editorial and formatting guidelines.

We are now performing detailed checks on your paper and will send you a checklist detailing our editorial and formatting requirements in about two weeks. Please do not upload the final materials and make any revisions until you receive this additional information from us.

To facilitate our work at this stage, it is important that we have a copy of the main text as a word file. If you could please send along a word version of this file as soon as possible, we would greatly appreciate it; please make sure to copy the NSMB account (cc'ed above).

Sincerely,

Dimitris Typas
Associate Editor
Nature Structural & Molecular Biology
ORCID: 0000-0002-8737-1319

Reviewer #1 (Remarks to the Author):

I think that this is an excellent manuscript that would be of interest to diverse readership of NSMB. In the revised version, all my concerns have been addressed and I recommend that this article is accepted for publication without any additional changes.

Reviewer #2 (Remarks to the Author):

In revising their manuscript, the authors adequately addressed all points raised by this reviewer.

Final Decision Letter:**Message** 16th Oct 2023

:

Dear Dr. Nudler,

We are now happy to accept your revised paper "General Transcription Factor from E. coli with a Distinct Mechanism of Action" for publication as an Article in Nature Structural & Molecular Biology.

As soon as your article is published, you can generate your shareable link by entering the DOI of your article here: http://authors.springernature.com/share. Corresponding authors will also receive an automated email with the shareable link

Your paper will be published online soon after we receive proof corrections and will appear in print in the next available issue. You can find out your date of online publication by contacting the production team shortly after sending your proof corrections. Content is published online weekly on Mondays and Thursdays, and the embargo is set at 16:00 London time (GMT)/11:00 am US Eastern time (EST) on the day of publication. Now is the

time to inform your Public Relations or Press Office about your paper, as they might be interested in promoting its publication. This will allow them time to prepare an accurate and satisfactory press release. Include your manuscript tracking number (NSMB-A47694B) and our journal name, which they will need when they contact our press office.

About one week before your paper is published online, we shall be distributing a press release to news organizations worldwide, which may very well include details of your work. We are happy for your institution or funding agency to prepare its own press release, but it must mention the embargo date and Nature Structural & Molecular Biology. If you or your Press Office have any enquiries in the meantime, please contact press@nature.com.

Please note that *Nature Structural & Molecular Biology* is a Transformative Journal (TJ). Authors may publish their research with us through the traditional subscription access route or make their paper immediately open access through payment of an article-processing charge (APC). Authors will not be required to make a final decision about access to their article until it has been accepted. Find out more about Transformative Journals <https://www.springernature.com/gp/open-research/transformative-journals>

Authors may need to take specific actions to achieve [compliance](https://www.springernature.com/gp/open-research/funding/policy-compliance-faqs) with funder and institutional open access mandates. If your research is supported by a funder that requires immediate open access (e.g. according to [Plan S principles](https://www.springernature.com/gp/open-research/plan-s-compliance)) then you should select the gold OA route, and we will direct you to the compliant route where possible. For authors selecting the subscription publication route, the journal's standard licensing terms will need to be accepted, including [self-archiving policies](https://www.springernature.com/gp/open-research/policies/journal-policies). Those licensing terms will supersede any other terms

that the author or any third party may assert apply to any version of the manuscript.

Sincerely,

Dimitris Typas
Associate Editor
Nature Structural & Molecular Biology
ORCID: 0000-0002-8737-1319
